# BALANCING PLASTICITY AND STABILITY WITH FAST AND SLOW SUCCESSOR FEATURES

## ABSTRACT

A hallmark of intelligence is the ability to adapt in non-stationary environments, yet deep Reinforcement Learning (RL) agents often struggle in such settings. Most prior studies introduce non-stationarity through abrupt shifts in features or dynamics, whereas real-world changes might be more gradual, reflecting naturalistic continual drift in the underlying dynamics. This may have important implications for studies on the "stability versus plasticity dilemma" in RL, since abrupt changes in the task may necessitate more plasticity than real-world situations actually would demand. To address these concerns, we modify existing 3D Miniworld and MuJoCo environments to incorporate naturalistic, continual non-stationary changes, and use them to identify whether poor performance in RL systems arises from a loss of plasticity or stability. We find that in these settings, methods that preserve stability, such as synaptic consolidation, achieve better performance than those focused on plasticity, such as resetting of a subset of the parameters. Motivated by this finding, and prior evidence that successor features (SFs) reduce interference in non-stationary settings, we investigate whether SFs provide a better target than Q-values for consolidation. Across both environments, we find that applying a neuro-inspired synaptic consolidation mechanism to SFs rather than Q-values yields superior performance on the naturalistic, continual changing MuJoCo tasks. Furthermore, we find that consolidation is most effective when SFs are stabilized across multiple timescales, as different timescales capture complementary aspects of the gradually changing environment. Together, these results show that stability may be more important in continual learning settings when abrupt changes in tasks do not occur. Moreover, to enhance stability, multi-timescale consolidation of predictive representations is an effective approach.

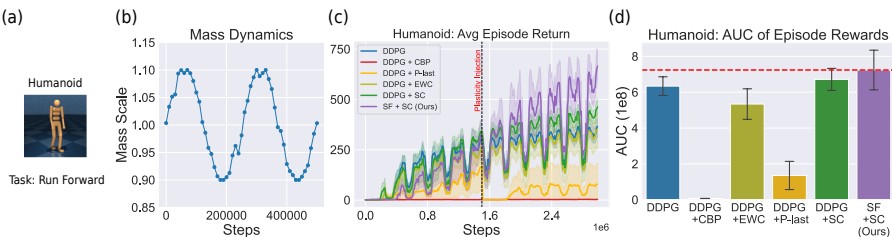

Figure 1: Motivating stability-plasticity tradeoffs in naturalistic, continually non-stationary RL where the environment evolves gradually, rather than abruptly. To illustrate, we show **(a)** the Humanoid running forward task and **(b)** an example noisy sine function used to generate smooth changes in its mass. **(c)** Average episode return plot and **(d)** Area under the curve (AUC) show that stability-preserving methods (EWC, SC) outperform purely plastic ones (CBP, P-last), with further gains from consolidating SFs (SF+SC, purple) rather than Q-values (DDPG+SC, green). Plasticity injection for DDPG+P-last (yellow) was performed halfway through the training.

## 1 INTRODUCTION

Events in the real world are often constantly evolving. Humans and animals must therefore adapt in environments where the underlying dynamics shift naturally and continually. In contrast, many

continual learning studies in Artificial Intelligence (AI) focus on abrupt, task-boundary changes, where the features or dynamics across tasks differ substantially. Standard RL techniques, such as Q-learning, struggle under such conditions and often suffer from catastrophic forgetting (McCloskey & Cohen, 1989; French, 1999). Developing methods that enable deep RL agents to learn effectively in naturalistic, continually changing environments remain a major goal in AI research (Khetarpal et al., 2022; Abel et al., 2023; Silver & Sutton, 2025).

While early work in supervised continual learning emphasized stability (i.e., the ability to retain previously acquired knowledge and prevent catastrophic forgetting) (Kirkpatrick et al., 2017; Zenke et al., 2017), RL poses unique challenges, since changing policies alter the samples an agent encounters, compounding environmental non-stationarity. In RL, Atari (Bellemare et al., 2013) has emerged as one of the standard testbeds for sequential task learning, where stability-focused methods such as Elastic Weight Consolidation (EWC) (Kirkpatrick et al., 2017) and replay (Rolnick et al., 2019) became dominant strategies. More recently, studies have shifted towards the complementary issue of plasticity (i.e., the capacity to rapidly adapt to new experiences), using either sequential Atari tasks (Abbas et al., 2023) or artificial tasks in MuJoCo created by randomly sampling friction coefficients per task (Dohare et al., 2024). Stability has mostly been studied in sequential multi-task settings and plasticity in single-task dynamics, but real-world environments rarely fit either case, as naturalistic, continual non-stationarity appears as a single task while still creating a stream of shifting sub-tasks.

Despite these advances, it remains unclear how stability and plasticity trade off in more real-world-like environments that undergo naturalistic, continual non-stationarity, where agents must adapt to naturalistic and continual changes without explicit task boundaries. A natural way to study this problem is to develop environments with naturalistic, continually evolving dynamics and compare algorithms that are task agnostic, and either enhance plasticity (e.g., parameter resets (Nikishin et al., 2022; 2023; Sokar et al., 2023; Dohare et al., 2024; Lee et al., 2024)) or preserve stability (e.g., consolidation that either protects important parameters (Kirkpatrick et al., 2017) or allow learning across multiple timescales (Kaplanis et al., 2018; 2019; Anand & Precup, 2023)). While other approaches exist, such as replay-based methods (Riemer et al., 2018; Rolnick et al., 2019; Caccia et al., 2023), they are less suited to our setting since their benefits rely on storing and mixing past and recent samples, which is problematic when no clear task boundaries exist. More broadly, most prior approaches tackle the stability-plasticity trade-off at the level of Q-values or policies, leaving the role of representations largely understudied.

In this paper, we explore whether alternative representations could help balance plasticity and stability. Successor Features (SFs) (Barreto et al., 2017; Borsa et al., 2018; Chua et al., 2024) are a good candidate, as they capture predictive structures that enable rapid transfer across tasks with shared dynamics while also reducing interference. We investigate whether predictive representations, like SFs, can serve as a basis for achieving both plasticity and stability under naturalistic, continual non-stationarity.

We perform experiments in two environments with evolving dynamics: a slippery four rooms environment, where actions are occasionally replaced, and MuJoCo control tasks, where the embodiment's mass changes. Slippery dynamics capture variability in action outcomes (e.g., wet or icy ground), while mass changes reflect natural shifts in body weight. Non-stationarity is induced by sampling these variables from a noisy sine function, similar to prior work on continuous dynamics (Xie et al., 2020). We test whether poor performance stems from instability or loss of plasticity by comparing base models (DQN or DDPG) with variants that inject plasticity via parameter re-initialization, or preserve stability through consolidation. Our analyses show that stability is the critical bottleneck, with synaptic consolidation (Benna & Fusi, 2016; Kaplanis et al., 2018) emerging as the most effective mechanism. Building on this, we combine SFs with synaptic consolidation to consolidate predictive representations across multiple timescales, offering a principled balance between plasticity and stability. Finally, we analyze SFs across timescales to reveal how stability and plasticity are distributed over temporal dimensions.

## 2 RELATED WORK

Our work builds upon prior studies of stability or plasticity in RL. Early work to mitigate forgetting emphasized stability, introducing methods that use importance measures such as Fisher information

to protect parameters critical for previous tasks (Kirkpatrick et al., 2017; Schwarz et al., 2018), manipulating replay mechanisms (Rolnick et al., 2019; Riemer et al., 2018; Kaplanis et al., 2004; Caccia et al., 2023), augmenting architectures (Powers et al., 2022), or employing consolidation systems that maintain multiple sets of parameters updated at different timescales (Kaplanis et al., 2018; 2019; Anand & Precup, 2023). Several approaches explicitly rely on task information, for example by using task boundaries to trigger consolidation (Kirkpatrick et al., 2017) or distillation phases (Schwarz et al., 2018). Other approaches that are described as task-agnostic, but nevertheless rely on auxiliary mechanisms such as recency tracking by separating 'new' or 'replay' samples (Rolnick et al., 2019), or the use of drift detection mechanisms that trigger architecture adaptations (Powers et al., 2022). These assumptions are problematic in environments that evolve naturally and continually, where there are no discrete task boundaries to detect and the very notion of "new" versus "old" experiences becomes ill-defined.

More recently, studies in continual RL have shifted attention to the problem of loss of plasticity. By analyzing neural activities, effective rank of the representations, and gradient dynamics during training, proposed mitigation strategies have focused on modifying the activation functions or optimizers (Ben-Iwhiwhu et al., 2022; Abbas et al., 2023), regularizing the parameters using weight decay or normalization (Lyle et al., 2024), and, more commonly, injecting plasticity by resetting subsets of network parameters such as the last few layers (Nikishin et al., 2022; 2023) or the ones that are least active (Sokar et al., 2023; Dohare et al., 2024). However, most of these approaches have been evaluated only in discrete or single-task settings, or under non-stationarity that is abrupt rather than naturally and continually.

Among these prior approaches, our study is most closely related to consolidation-based approaches (Kirkpatrick et al., 2017; Schwarz et al., 2018; Kaplanis et al., 2018) and to recent efforts examining loss of plasticity in deep RL (Nikishin et al., 2023; Dohare et al., 2024) as they do not require explicit or implicit task statistics. However, they have yet to be evaluated under naturalistic, continually evolving settings, and remain limited to discrete tasks or single-task settings. Moreover, whether such approaches are effective when applied to learned representations, rather than Q-values or policies, remain unclear. In this work, we address these limitations by analyzing stability and plasticity under naturalistic, continual changes, and by proposing a synaptic consolidation system with SFs, that consolidates representations across multiple timescales.

## 3 PRELIMINARIES

### 3.1 REINFORCEMENT LEARNING

We consider the RL setting as a Markov Decision Process, defined by a tuple $(S, A, p, r, \gamma)$, where $\mathcal{S}$ is the set of states, $\mathcal{A}$ is the set of actions, $r : S \to \mathbb{R}$ is the reward function, $p : \mathcal{S} \times \mathcal{A} \to [0, 1]$ is the transition probability function and the discount factor $\gamma \in [0, 1)$ which determines the importance of immediate and future rewards (Sutton & Barto, 2018).

At each time step $t$, the agent observes state $S_t \in \mathcal{S}$ and takes an action $A_t \in \mathcal{A}$ sampled from a policy $\pi : \mathcal{S} \times \mathcal{A} \to [0, 1]$, resulting to the transition of next state $S_{t+1}$ with probability $p(S_{t+1} \mid S_t, A_t)$ and the reward $R_{t+1}$.

### 3.2 SUCCESSOR FEATURES

SFs are defined via a decomposition of the state-action value function (i.e. the expected return), $Q$, into the reward function and a representation of expected features occupancy for each state $S_t$ and action $A_t$ of time step $t$:

$$Q(S_t, A_t, \boldsymbol{w}) = \psi(S_t, A_t, \boldsymbol{w})^\top \boldsymbol{w} \tag{1}$$

where $\psi \in \mathbb{R}^n$ are the SFs that capture expected feature occupancy and $\boldsymbol{w} \in \mathbb{R}^n$ is a vector of the task encoding, which can be considered a representation of the reward function (Borsa et al., 2018).

Canonically, the SFs for a state-action pair $(s, a)$ under a policy $\pi$ are defined as:

$$\psi^\pi(s, a) \equiv \mathrm{E}^\pi \left[ \sum_{i=t}^{\infty} \gamma^{i-t} \phi_{i+1} \mid S_t = s, A_t = a \right] \tag{2}$$

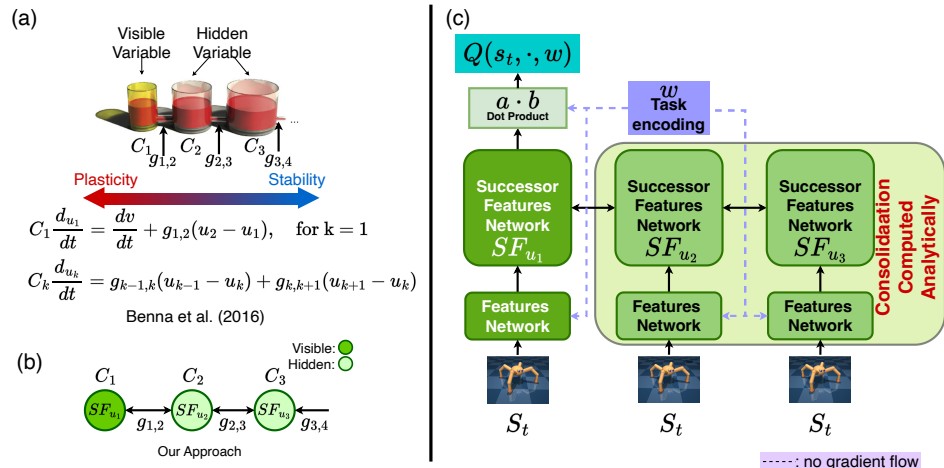

Figure 2: **a:** Neuro-inspired synaptic consolidation model adapted from (Benna & Fusi, 2016). The visible variable, $u_1$, represents the synaptic efficacy $v$, while downstream hidden variables $u_2, u_3, ...$ interact bidirectionally across timescales, with beaker capacities $C_1 < C_2 < ..., < C_K$ and tube widths representing flow strength $g_{1,2} > g_{2,3} > ..., > g_{K,K+1}$ controlling the rate of interaction between the variables. Together, the beaker sizes ($C_k$) and flow strength ($g_{k,k+1}$) govern the effective timescales of plasticity and stability. **b:** The synaptic efficacy $v$ is replaced by the parameters of SFs, thus allowing SFs to be learned across different timescales. **c:** Our architectural design. See section 4 for more details on training the system.

where $\phi \in \mathbb{R}^n$ is a set of basis features (Barreto et al., 2017).

However, as shown by Borsa et al. (2018), we can treat the task encoding vector $\boldsymbol{w}$ as a way to encode policy $\pi$, resulting in *Universal SFs*, $\psi(s, a, \boldsymbol{w})$. The task encoding vector $\boldsymbol{w}$ can also be related directly to the rewards themselves via the underlying basis features ($\phi$):

$$R_{t+1} = \phi(S_{t+1})^\top \boldsymbol{w} \tag{3}$$

Among the various SF learning techniques, our approach builds on Simple SFs (Chua et al., 2024), which can be learned during task engagement, and do not require additional auxiliary losses or pre-training. Both the SFs $\psi$ and the task encoding vector $w$ are learned by optimizing the following two losses:

$$L_w = \frac{1}{2} \left\| R_{t+1} - \overline{\phi}(S_{t+1})^\top \boldsymbol{w} \right\|^2 \tag{4}$$

$$L_\psi = \frac{1}{2} \left\| \hat{y} - \psi(S_t, A_t, \boldsymbol{w})^\top \boldsymbol{w} \right\|^2 \tag{5}$$

where $\overline{\phi}(S_{t+1})$ is the L2-normalized output from the feature network (Figure 25 in Appendix M), and is treated as a constant in Eq. 4 by using a stop-gradient operator. $\hat{y}$ is the bootstrapped target:

$$\hat{y} = R_{t+1} + \gamma \max_{a'} \psi(S_{t+1}, a', \boldsymbol{w})^\top \boldsymbol{w} \tag{6}$$

### 3.3 NEURO-INSPIRED SYNAPTIC CONSOLIDATION MECHANISM

In this work, we revisit the Synaptic Consolidation mechanism (SC) (Benna & Fusi, 2016) which has been previously adapted to deep RL (Kaplanis et al., 2018; 2019). Despite these adaptations, SC remains far less studied than Systems Consolidation (McClelland et al., 1995) in AI. We revisit SC because (i) it can be learned without explicit or implicit task statistics, (ii) it generalizes beyond dual fast/slow schemes by supporting multiple timescales, (iii) its linear-chain formulation provides a smooth, principled balance of stability and plasticity without ad-hoc mechanisms, and (iv) it has already been shown to be effective in deep RL (Kaplanis et al., 2018; 2019). These properties make SC particularly well-suited to SFs, allowing predictive representations to be preserved during continual learning.

To make this concrete, we now outline the SC mechanism — originally proposed to model how synaptic strength stabilizes over time (Benna & Fusi, 2016) — which can be understood as a chain of $K$ interacting variables, $u_1, u_2, \ldots, u_K$, each associated with a capacity $C_k \in \mathbb{Z}+$. The first variable $u_1$, corresponds to visible synaptic efficacy $v$, i.e., the strength of the connection between two neurons, and is the most plastic component (see Figure 2a. for a schematic of this system). Its dynamics are:

$$C_1 \frac{d_{u_1}}{dt} = \frac{dv}{dt} + g_{1,2}(u_2 - u_1), \quad \text{for k = 1} \tag{7}$$

where $g_{1,2} \in \mathbb{R}$ determines the flow strength between $u_1$ and $u_2$.

Interior variables $u_k$, interact bidirectionally with their two neighbors:

$$C_k \frac{d_{u_k}}{dt} = g_{k-1,k}(u_{k-1} - u_k) + g_{k,k+1}(u_{k+1} - u_k) \quad \text{for k = 2,3, \ldots, K-1} \tag{8}$$

with $g_{k-1,k}, g_{k,k+1} \in \mathbb{R}$. Finally, the last variable $u_K$, has no downstream neighbor. Thus, setting $u_{K+1} \leftarrow 0$ produces a natural leak term that induces decay:

$$C_K \frac{d_{u_K}}{dt} = g_{K-1,K}(u_{K-1} - u_K) + g_{K,K+1}(-u_K) \tag{9}$$

Together, the capacity $C_k$ and the flow strength $g_{k,k+1}$ define the continuous timescales of plasticity and stability of each variable $u_k$. To implement these dynamics in RL, which operates in discrete steps, we discretize them with Euler's method.

## 4 LEARNING SUCCESSOR FEATURES WITH SYNAPTIC CONSOLIDATION

In line with prior work which discretize synaptic consolidation by replacing synaptic efficacy $v$ with the parameters of a Q-value function (Kaplanis et al., 2018) or a policy (Kaplanis et al., 2019), we instead apply synaptic consolidation to the parameters of the SFs. Specifically, each variable $u_k$ is mapped to the corresponding SF parameters $\theta_k \in \mathbb{R}^n$, yielding $\psi_{u_k} = \psi_{\theta_{u_k}} \in \mathbb{R}^n$, where $\psi$ denotes the SFs. For brevity, we will write $\psi_{u_k}$ to denote $\psi_{\theta_{u_k}}$. We next derive the learning rules under this discretization using Euler's method.

Let $\eta_k = \Delta t / C_k$. There will be two learning phases ($t + \frac{1}{2}$ and $t + 1$) for the most plastic variable, SF $\psi_{u_1}$. At phase $t + \frac{1}{2}$, SF $\psi_{u_1}$ is learned via optimizing Q-SF-TD loss $L_\psi$ (Eq. 5):

$$\psi_{u_1}^{t+\frac{1}{2}} = \psi_{u_1}^t - \alpha \nabla_{\psi_{u_1}} L_{\psi_{u_1}} \tag{10}$$

where $\alpha \in \mathbb{R}$ is the learning rate. At the second phase, $t + 1$, we update $\psi_{u_1}$ and the rest of the SFs variable ($\psi_{u_2}, \psi_{u_3}, \ldots, \psi_{u_K}$) using the Euler update. For the first variable $\psi_{u_1}$:

$$\psi_{u_1}^{t+1} = \psi_{u_1}^{t+\frac{1}{2}} + \eta_1 \left[ g_{1,2} \left( \psi_{u_2}^t - \psi_{u_1}^{t+\frac{1}{2}} \right) \right] \tag{11}$$

For the interior variables $k = 2, \ldots, K - 1$:

$$\psi_{u_k}^{t+1} = \psi_{u_k}^t + \eta_k \left[ g_{k-1,k}(\psi_{u_{k-1}}^t - \psi_{u_k}^t) + g_{k,k+1}(\psi_{u_{k+1}}^t - \psi_{u_k}^t) \right] \tag{12}$$

For the last variable $K$:

$$\psi_{u_K}^{t+1} = \psi_{u_K}^t + \eta_K \left[ g_{K-1,K}(\psi_{u_{K-1}}^t - \psi_{u_K}^t) - g_{K,K+1}(\psi_{u_K}^t) \right] \tag{13}$$

In our experiments, we set $\Delta t = 1$, $C_k = 2^k$ and used a constant flow strength between variables, i.e $g_{k-1,k} = g_{1,2}$ for all $k$. We apply a global scaling factor $s \in \mathbb{Z}_+$ that multiplies all capacities ($C_k \rightarrow s \cdot C_k$), thereby ensuring the timescales extend far beyond the total training steps and keeping the number of variables required small. See Figure 2b and 2c on how the synaptic consolidation system is adapted for SFs.

We provide a pseudocode of the algorithm in Appendix C. It is important that these updates (Eqs. 11, 12 and 13) are performed using Stochastic Gradient Descent (SGD) rather than adaptive approaches like Adaptive Moment Estimation (Adam, (Kingma & Ba, 2014)), which do not preserve the timescales information. We provide a proof sketch in Appendix B supporting this claim.

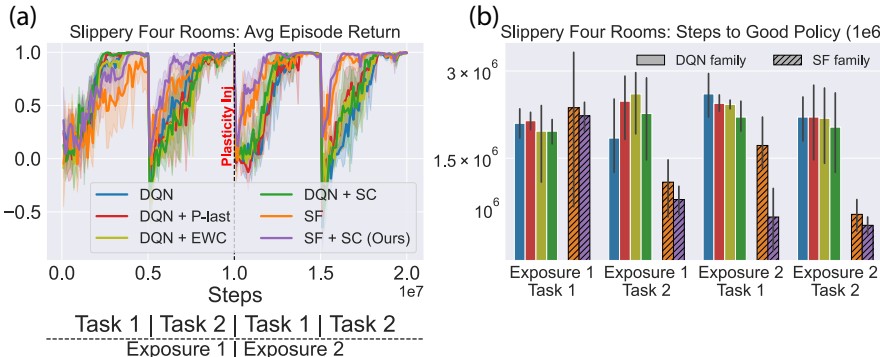

Figure 3: Results from Slippery Four Rooms with naturalistic, continual evolving slip dynamics that randomly replace actions. **(a):** Average return across two sequential tasks (Task 1 and 2), each repeated twice (Exposure 1 and 2). In DQN+P-last (yellow), plasticity injection is applied midway through training by randomly re-initializing the last layer's parameters. **(b):** Steps to reach a predefined performance threshold (fewer is better). Overall, stability-preserving methods (EWC, SC) outperform plastic ones (P-last), with further gains from combining SFs with SC (SF+SC).

## 5 EXPERIMENTAL RESULTS

In this study, we consider two environments. The first is a slippery variant of the 3D Four Rooms environments, adapted from Chua et al. (2024), which mimics conditions such as walking on wet or icy surfaces. In this environment, the 'slippery' event refers to the agent's action being randomly replaced by an alternative, based on a probability value sampled from a noisy sine function to simulate naturalistic, continual dynamics shifts (Figure 9 in Appendix D). The agent alternates between two tasks, where in the first task it receives a reward of +1 for reaching the green box and -1 for reaching the yellow box, and in the second task the rewards are reversed. The agent cycles through this two-task sequence twice and only receives egocentric pixel observations (see Figure 4).

The second environment is the MuJoCo suite (Todorov et al., 2012), using the Deep-Mind Control Suite (DMC) (Tunyasuvunakool et al., 2020), which provides an accessible framework for modifying dynamics of the embodiments. We focus on four embodiments, Half-cheetah, Walker, Quadruped and Humanoid, ordered by increasing complexity in terms of their observation and action spaces. To simulate naturalistic, continual dynamics shifts, at every ten episodes, we perturbed the embodiment's mass by sampling from a noisy sine function. In this task, agents are rewarded for running forward.

For base models, we use Double Deep Q-Network (DQN) (Van Hasselt et al., 2016) for Slippery Four Rooms environment and the Deterministic Policy Gradient (DDPG) algorithm (Silver et al., 2014) with twin critics for MuJoCo. For SFs, we use Simple SFs (Chua et al., 2024) which can be learned without auxiliary losses. We selected these models due to their flexibility, which makes extensions with plasticity injections or synaptic consolidation mechanisms rather straightforward.

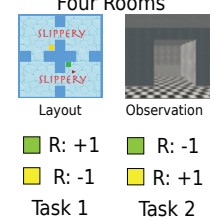

Figure 4: Slippery Four Rooms

We compared against baselines that do not require task statistics. For plasticity, we reset subsets of parameters—last layer (DQN+P-last, DDPG+P-last) (Nikishin et al., 2023) or least used via continual backprop (DDPG+CBP) (Dohare et al., 2024), the latter more effective in MuJoCo. For stability, we used online Elastic Weight Consolidation (EWC) (Schwarz et al., 2018) for Q-values and SFs (DQN+EWC, DDPG+EWC, SF+EWC). We also included synaptic consolidation (SC, Figure 2) with Q-value (DQN+SC, DDPG+SC) and SF variants (SF+SC). Results are averaged over 5 seeds.

Our experiments are designed to address the following questions:

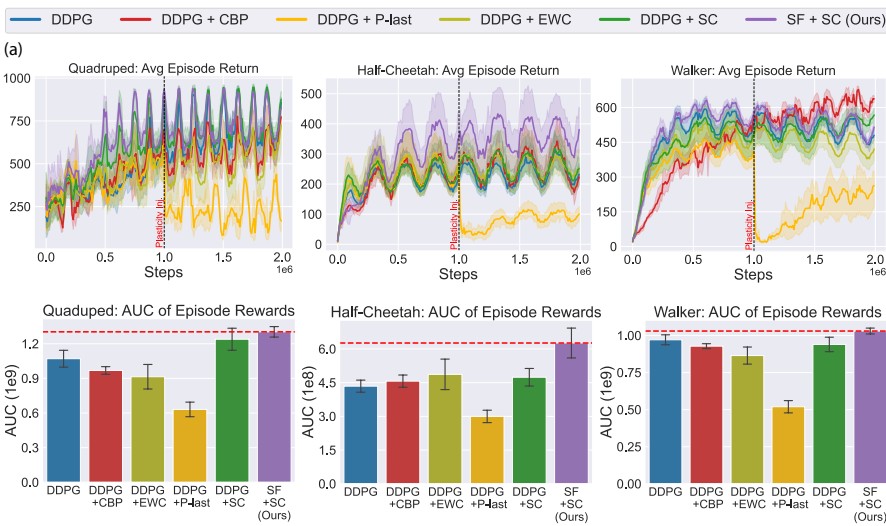

Figure 5: Results from MuJoCo, where agent embodiments undergo continuous mass changes during training. **(a):** Average episode return. Plasticity injection for DDPG+P-last (yellow) was performed halfway through the training. **(b):** Area under the Curve (AUC) of returns in (a). Together, the plots show that stability-preserving methods (EWC and SC) outperform plastic ones (CBP and P-last), with further gains achieved by combining synaptic consolidation with SFs (SF+SC). Results for Humanoid can be seen in Figure 1(c-d).

1. When agents undergo naturalistic, continual non-stationary shifts, is the primary bottleneck one of plasticity or stability? If stability is the limiting factor, which consolidation mechanism is more effective? Elastic Weight Consolidation (EWC) or Synaptic Consolidation (SC)?

2. Second, is it more effective to consolidate parameters of Q-value or Successor Features?

## 5.1 STABILITY AS THE BOTTLENECK: SYNAPTIC CONSOLIDATION OUTPERFORMS EWC

To study the question of plasticity versus stability, we first evaluated the performance in the slippery Four Rooms environment using DQN, along with its plasticity injection variants (P-last) and stability-preserving variants (EWC and SC). The results in Figure 3 show that stability-preserving models (DQN + EWC and DQN + SC) consistently outperformed the plasticity-injection model (DQN + P-last), with synaptic consolidation (DQN + SC) achieving higher learning efficiency.

We next evaluated performance in the MuJoCo suite using DDPG, together with its two plasticity-injection variants (P-last and CBP) and stability-preserving variants (EWC and SC). The results in Figure 5 show that stability-preserving models (DDPG + EWC and DDPG + SC) consistently outperformed the plasticity-injection model (DDPG + P-last and DDPG + CBP). Once again, synaptic consolidation (DDPG + SC) achieved higher learning efficiency, particularly in the more complex embodiments, Humanoid (Figure 1) and Quadruped (Figure 5a and b).

Both sets of results indicate that *stability is the primary bottleneck* as agents lacking stability fail to learn effectively in the naturalistic, continual non-stationary environments considered in this study. The results further showed that the synaptic consolidation (SC) mechanism is more effective than Elastic Weight Consolidation (EWC).

Given this finding, the natural next question is, what exactly should be stabilized: the Q-value parameters themselves, or the parameters of the underlying representations such as SFs?

## 5.2 WHAT SHOULD BE STABILIZED: Q-VALUES OR SUCCESSOR FEATURES?

As many stability preserving methods focus on stabilizing parameters of Q-value functions, we investigate if SFs could be a better target for consolidation. To address this, we evaluated a SFs variant

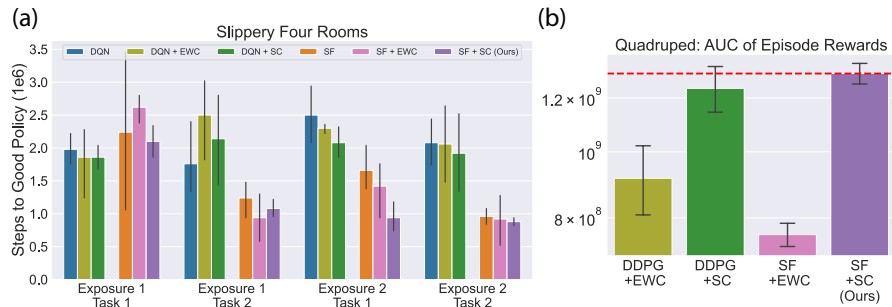

Figure 6: Comparison of consolidating Q-values vs. Successor Features (SFs) using Elastic Weight Consolidation (EWC) or Synaptic Consolidation (SC). **(a):** Learning efficiency in Slippery Four Rooms, measured as steps to reach a reasonable policy (lower is better). **(b):** Area under the curve (AUC) of episode rewards in Quadruped (higher is better). Overall, combining SFs with SC is most effective. Additional MuJoCo results are in Appendix H.2.

combined with synaptic consolidation (SF + SC) in both slippery Four Rooms and the MuJoCo suite. The results for both slippery Four Rooms (Figure 3) and the MuJoCo suite (Figures 1 and 5) showed that SF + SC consistently improves performance compared to Q-value based consolidation. We also compared with EWC, and the results for slippery Four Rooms and Quadruped are shown in Figure 6. These comparisons reveal that while SC is more effective than EWC overall, only the combination of SFs with SC yields consistently strong performance.

# 6 ANALYZING MULTI-TIMESCALE CONTRIBUTIONS

To gain insights into why combining SFs and Synaptic Consolidation yields an effective model, we perform ablation studies by varying the number of consolidation variables, and we complement this with a cross-attention (Dosovitskiy et al., 2020) analysis to analyze the relative contributions of individual variables.

## 6.1 DO FAST OR SLOW TIMESCALE VARIABLES MATTER MORE FOR LEARNING?

In this analysis, we varied over the number of consolidation variables (3, 6, 8, or 9), with fewer variables yielding greater plasticity, and more variables yielding greater stability. The aim was to assess whether the inclusion of slower timescale variables improve policy learning.

Figure 7 illustrates the results of the Humanoid, Quadruped, and Half-Cheetah embodiments. We found that using six or more timescale variables leads to better learning performance. For the slippery four rooms environment (Figure 19 in Appendix 6.1), we observed a similar trend: using six or nine consolidation variables leads to better performance. Together, these results demonstrated that preserving stability is crucial in naturalistic, continual non-stationary settings.

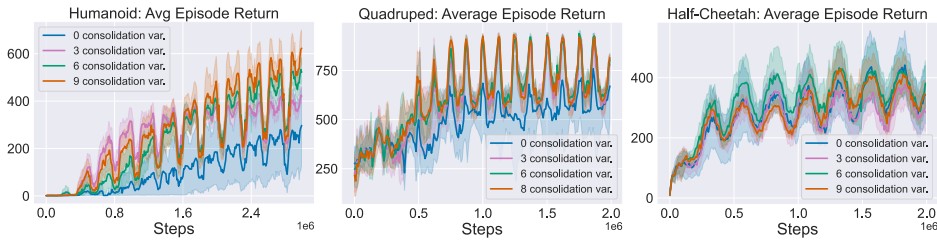

Figure 7: Analysis of timescales in MuJoCo. More consolidation variables (6–9) improve learning efficiency, highlighting the benefit of slower timescales. Zero variables correspond to the Simple SF agent. See Appendix J for results in the Slippery Four Rooms environment.

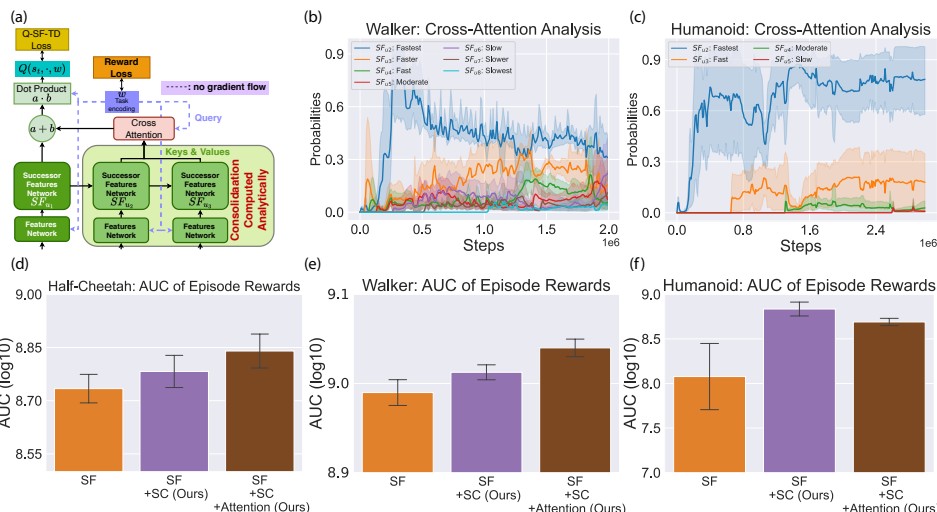

Figure 8: Cross-Attention analysis of individual consolidations, replacing memory recall via back-flow. **(a):** Implementation design. **(b-c):** Attention probabilities over consolidation variables, where higher probability indicates greater contribution to policy learning. **(d-f):** Area under the curve (AUC) of average episode returns: simpler embodiments (Half-Cheetah, Walker) benefited, while more complex ones (Humanoid) performed worse. Full results are provided in Appendix L.

## 6.2 WHAT DOES CROSS-ATTENTION REVEAL ABOUT THE CONTRIBUTIONS OF CONSOLIDATION VARIABLES?

While varying over the number of consolidation variables provides a coarse measure of their utility, it does not reveal which specific variables contribute most. At the same time, using a cross-attention mechanism also allow us to bypass the need for information to propagate gradually via the flow strength $(g_{k,k+1})$, instead providing an instant readout from all the consolidation variables (see Figure 8a).

We adapt the canonical cross-attention mechanism by letting the task encoding vector $w$ serves as the query, while the SF consolidation variables (excluding the most plastic $SF_{u_1}$) serve as keys and values. To construct more discriminative representations, we subtracted each variable from its faster neighbor (eg. $SF_{u_k} = SF_{u_k} - SF_{u_{k-1}}$). The keys and values were layer-normalized before projected through learnable weights $(W_{\text{Keys}}, W_{\text{Values}})$, while the query vector $w$ is projected through $W_{\text{Query}}$. Attention scores were computed via the query-keys multiplication, followed by softmax activation, which was then multiplied by the Values to produce a weighted sum, thus integrating information across timescales (see Appendix K for more details). Since the deeper SF consolidation variables $(SF_{u_2}, SF_{u_3}, \ldots)$ were computed analytically and not part of the computational graph, we applied a reparameterization trick (inspired by Kingma & Welling (2013)) to enable joint training of the cross-attention mechanism with the most plastic variable $(SF_{u_1})$ using the Q-SF-TD loss (Eq. 5).

The number of consolidation variables were kept consistent with those used in the SF + SC model, except for Humanoid, where it was reduced to improve learning efficiency. The softmax probabilities for Walker and Humanoid (Figure 8b and c) showed that the faster timescale variables, particularly $SF_{u_2}$ and $SF_{u_3}$, received the highest attention. However, the slower variables also contributed, more prominently in Walker (Figure 8b), as well as in Half-Cheetah and Quadruped (Figure 23 in Appendix L). Furthermore, we also observed that memory recall via attention mechanism led to improved performance in Half-Cheetah and Walker, but not in Humanoid or Quadruped (Figure 23), likely due to the additional complexity of learning the attention mechanism in more complex embodiments. Together, these findings suggest that fast timescales drive most learning, while slower ones provide complementary stability that aids performance in simpler embodiments.

## 7 DISCUSSION

In this work, we introduced naturalistic, continual non-stationarity environments to study stability-plasticity trade-off in deep RL, and showed that stability is the primary bottleneck. By adapting a neuro-inspired synaptic consolidation mechanism with SFs, we developed a multi-timescale system that balances stability and plasticity through predictive representations. By consistently outperforming its counterparts, our approach shows that combining synaptic consolidation with fast and slow SFs is key to robust learning under naturalistic, continual non-stationarity. Future work could build on this to develop more effective attention mechanisms to further improve memory recall and consolidation.

## 8 REPRODUCIBILITY STATEMENT

We implemented our experiments primarily in JAX (for the Slippery Four Rooms environment) and PyTorch (for the MuJoCo tasks). Appendix B contains a mathematical proof showing why the consolidation system requires updates via Stochastic Gradient Descent rather than Adaptive Moment Estimation (Adam). The pseudocode of our algorithm is provided in Appendix C. Appendices D and E contain all the details of all the environments (Slippery Four Rooms environment and MuJoCo suite) setup. Full architectural details of our models, including hyperparameters, are provided in Appendix M, along with implementation specifics of the baseline models. Software packages, dependencies, and computational resources are listed in Appendix N. All experiments were conducted on Nvidia A100 GPUs, across five random seeds, with training runs completing in under two days. Results are presented as mean and standard deviation across seeds.

## 9 ETHICS STATEMENT

This work advances the scientific study of continual reinforcement learning by investigating the stability-plasticity tradeoffs in naturalistic, continual non-stationary environments. All experiments are conducted in simulated environments (Four Rooms, MuJoCo) and do not involve human or animal subjects. Overall, our work is conceptual and aims to contribute to safer and more reliable adaptive AI, as well as insights relevant to neuroscience. We used a large language model (ChatGPT) solely for grammar correction and improving clarity of wording. All ideas, analyses, and results are original to the authors.

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

## A  APPENDIX

This supplementary section provides detailed insights and additional information that supports the findings and methodology discussed in the main paper. Below is a brief overview of what each section contains:

| **Appendix Section** |
| --- |
| Appendix B: Proof Sketch on preserving timescales with SGD |
| Appendix C: Pseudocode Implementation |
| Appendix D: Environments |
| Appendix E: Experimental Details |
| Appendix F: Plasticity-Stability Analysis |
| Appendix G: Schematic of Synaptic Consolidation for Q-values vs. Successor Features |
| Appendix H: Q-values vs. SFs, Elastic Weight Consolidation vs Synaptic Consolidation Comparison |
| Appendix I: Q-values vs. SFs, With and Without Synaptic Consolidation Comparison |
| Appendix J: Analysis of Fast and Slow Timescale Variables |
| Appendix K: Architecture for recalling consolidated Successor Features using Cross-Attention |
| Appendix L: Cross-Attention Analysis of Fast and Slow Timescale Variables |
| Appendix M: Agents |
| Appendix N: Implementation details |
| Appendix O: Computational Complexity |

## B  PROOF SKETCH ON PRESERVING TIMESCALES WITH SGD

In this section, using mathematical analysis, we provide the intuition why the learning of the multiple-timescales Successor Features (SFs) must be done using Stochastic Gradient Descent (SGD) and not the commonly used Adaptive moment estimation (Adam) (Kingma & Ba, 2014).

For the sake of brevity, we consider a tabular Reinforcement Learning setting where the SFs $\psi(s,a)$ are not parameterised and depends only on a state-action pair $(s,a)$:

$$\psi(s,a) = \mathbb{E}_\pi\left[\sum_{t=0}^{\infty} \gamma^t \phi(s_t, a_t) \mid s_0 = s, a_0 = a\right] \tag{14}$$

where $\gamma \in [0,1]$ is the discount factor and $\phi$ is the basis features. Without loss of generality, we consider an arbitrary consolidation system with $K \in \mathbb{Z}$ possible consolidation variables operating at $K$ possible timescales, where $K > 1$. Let $u_k$ be a variable and $\psi_{u_k} \in \mathbb{R}^n$ be the SFs operating at timescale $k \in (1, 2, ..., K)$. The terms $g_{k-1,k}/C_k, g_{k,k+1}/C_k$, where $C_k = 2^{k-1}$ and $g_{k,k+1} \propto 2^{-k-2}$ determines the overall timescales of learning the SFs $\psi_{u_k}$.

Recall that after applying the Euler's method to the continuous dynamics (section 3.3), we get the following update rules for the consolidation variables $(\psi_{u_1}, \psi_{u_2}, \dots, \psi_{u_K})$. Let $\eta_k = \Delta t / C_k$ and ignoring the first learning phase $t + 1/2$, at step $t + 1$, for the first variable $\psi_{u_1}$, we get:

$$\psi_{u_1}^{t+1} = \psi_{u_1}^{t+\frac{1}{2}} + \eta_1\big[g_{1,2}\big(\psi_{u_2}^t - \psi u_1^{t+\frac{1}{2}}\big)\big] \tag{15}$$

For the interior variables $k = 2, \dots, K - 1$:

$$\psi_{u_k}^{t+1} = \psi_{u_k}^t + \eta_k\big[g_{k-1,k}(\psi_{u_{k-1}^t} - \psi_{u_k}^t) + g_{k,k+1}(\psi_{u_{k+1}}^t - \psi_{u_k}^t)\big] \tag{16}$$

For the last variable $K$:

$$\psi_{u_K}^{t+1} = \psi_{u_K}^t + \eta_K\big[g_{K-1,K}(\psi_{u_{K-1}}^t - \psi_{u_K}^t) - g_{K,K+1}(\psi_{u_K}^t)\big] \tag{17}$$

Without loss of generality, we consider the case of updating $\psi_{u_1}$ in Eq. 15, where $u_1$ and $u_2$ represent the first and second consolidation variables. Let $\Delta t = 1$ and $\kappa_{1,2}$ be the timescale ratio $\frac{g_{1,2}}{C_1}$, the

update term for $\psi_{u_1}^{t+1}$, corresponding to:

$$\eta_1[g_{1,2}(\psi_{u_2}^t - \psi u_1^{t+\frac{1}{2}})] = \frac{g_{1,2}}{C_1}\big(\underbrace{\psi_{u_2}^t - \psi u_1^{t+\frac{1}{2}}}_{:=g_t}\big) \tag{18}$$

$$= \kappa_{1,2} \odot g_t \tag{19}$$

$$= \tilde{g}_t \tag{20}$$

In optimization, we usually use a learning rate, such as $\alpha \in \mathbb{R}$ to update the variables.

As we shall see in the proof sketch below, both $\kappa_{1,2} \in \mathbb{R}$ and $\alpha \in \mathbb{R}$ contribute to the effective learning rate. We will first present our analysis using SGD, followed by Adam.

### B.1 STOCHASTIC GRADIENT DESCENT (SGD)

Recall that the update rule for SGD at step $t$ is defined as:

$$\psi_{t+1}(s,a) \leftarrow \psi_t(s,a) - \alpha \odot \tilde{g}_t \tag{21}$$

**Proposition 1** *Optimizing Eq. 20 using Stochastic Gradient Descent (SGD) ensures that the gradient updates explicitly scales with $\alpha$, thus preserving the relative timescale information.*

*Proof.* Without loss of generality, let $\kappa \in \{\kappa_{1,2}, \kappa_{2,3}, ..., \kappa_{K,K+1}\}$:

$$\psi_{t+1}(s,a) \leftarrow \psi_t(s,a) - \alpha\tilde{g}_t \tag{22}$$

$$= \psi_t(s,a) - \alpha(\kappa \cdot g_t) \qquad \text{Sub } \tilde{g}_t = \kappa \cdot g_t \text{ from Eq.19} \tag{23}$$

$$= \psi_t(s,a) - (\alpha \cdot \kappa \cdot g_t) \tag{24}$$

It is then trivial to conclude that when the learning rate $\alpha = 1$, the effective learning rate $\alpha \cdot \kappa \cdot g_t = \kappa \cdot g_t$, thus preserving the relative scale of the updates, even when the timescale ratio $\kappa$ decreases as we move down the chain of dynamic variables due to the fact that $\kappa_{1,2} >> \kappa_{2,3} >> ..., \kappa_{K,K+1}$. $\square$

### B.2 ADAPTIVE MOMENT ESTIMATION (ADAM)

Recall that the update rule for Adam Kingma & Ba (2014) at step $t$ is defined as:

$$m_t \leftarrow \beta_1 \cdot m_{t-1} + (1 - \beta_1) \cdot \tilde{g}_t \qquad \text{First moment} \tag{25}$$

$$v_t \leftarrow \beta_2 \cdot v_{t-1} + (1 - \beta_2) \cdot \tilde{g}_t^2 \qquad \text{Second moment} \tag{26}$$

$$\hat{m}_t \leftarrow \frac{m_t}{(1 - \beta_1^t)} \qquad \text{Bias correction for first moment} \tag{27}$$

$$\hat{v}_t \leftarrow \frac{v_t}{(1 - \beta_2^t)} \qquad \text{Bias correction for second moment} \tag{28}$$

$$\psi_{t+1}(s,a) \leftarrow \psi_t(s,a) - \frac{\alpha}{\sqrt{\hat{v}_t} + \epsilon} \cdot \hat{m}_t \tag{29}$$

where $\frac{\alpha}{\sqrt{\hat{v}_t}+\epsilon}$ is the effective learning rate and $\tilde{g}_t$ is the update term as defined in Eq. 20.

**Proposition 2** *Optimizing Eq. 20 using Adam results in gradient updates to not preserve the relative timescale information.*

Let's focus our analysis on the effective learning rate $\frac{\alpha}{\sqrt{\hat{v}_t}+\epsilon}$ and once again, without the loss of generality, let $\kappa \in \{\kappa_{1,2}, \kappa_{2,3}, ..., \kappa_{K,K+1}\}$:

$$\frac{\alpha}{\sqrt{\hat{v}_t} + \epsilon} = \frac{\alpha}{\sqrt{\frac{v_t}{(1-\beta_2^t)}} + \epsilon} \qquad \text{Sub Eq. 28 into } \hat{v}_t$$

$$= \frac{\alpha}{\sqrt{\frac{\beta_2 \cdot v_{t-1} + (1-\beta_2) \cdot \tilde{g}_t^2}{(1-\beta_2^t)}} + \epsilon} \qquad \text{Sub Eq. 26 into } \hat{v}_t$$

$$= \frac{\alpha}{\sqrt{\frac{\beta_2 \cdot v_{t-1} + (1-\beta_2) \cdot (\kappa \cdot g_t)^2}{(1-\beta_2^t)}} + \epsilon} \qquad \text{Sub } \tilde{g}_t = \kappa \cdot g_t \text{ from Eq.20}$$

$$(30)$$

We can observe that when the learning rate $\alpha = 1$, the effective learning rate, $\frac{1}{\sqrt{\hat{v}_t} + \epsilon}$, increases as the timescale ratio variable $\kappa$ decreases. This is due to the fact that as we move down the chain of consolidation variables, we will get $\kappa_{1,2} >> \kappa_{2,3} >> ..., \kappa_{K,K+1}$. This implies that the timescale will no longer be preserved, as the relative scale of the updates will now become inversely proportional with respect to $\kappa$ rather than proportional to $\kappa$. $\square$

### B.3 CONCLUSION

These analyses demonstrate that learning the Successor Features (SFs), $\psi_{u_k}$, using Stochastic Gradient Descent (SGD) rather than Adam is critical for preserving the timescale information intrinsic to these features. The differential impact of SGD and Adam on the behavior of the updates highlights the importance of choosing an appropriate optimization strategy in RL settings that require maintenance of structured timescale information.

## C PSEUDOCODE IMPLEMENTATION

---
**Algorithm 1** Learning Successor Features with Synaptic Consolidation

---
1: Determine the number of consolidation variables $(\psi_{u_2}, \psi_{u_3}, \ldots)$
2: Initialize task encoding vector $\boldsymbol{w}$
3: Initialize SF $\psi_\theta$ network, SF $\overline{\psi_\theta}$ target network
4: Set $\psi_{\theta_{u_1}} \leftarrow \psi_\theta$
5: Copy $\theta_{u_1}$ to the networks of the consolidation variables (e.g., $\psi_{\theta_{u_2}} \leftarrow \theta_{u_1}, \psi_{\theta_{u_3}} \leftarrow \theta_{u_1}, \ldots$)
6: **for** $t := 1, \text{T}$ **do**
7:     Receive observation $S_t$ from environment
8:     $A_t \leftarrow \epsilon$-greedy using $Q(S_t, \cdot \mid \boldsymbol{w})$
9:     Send $A_t$ to receive $S_{t+1}$ and $R_{t+1}$ from environment
10:     $a' \in \operatorname*{argmax}_b \overline{\psi_\theta}(S_{t+1}, b, \boldsymbol{w})^\top \boldsymbol{w}$
11:     $\hat{y} = R_{t+1} + \gamma \overline{\psi_\theta}(S_{t+1}, a', \boldsymbol{w})^\top \boldsymbol{w}$
12:     $\phi \leftarrow$ L2 normalized output from the encoder of SF $\psi$ network
13:     $loss_{\psi_\theta} = (\psi_\theta(S_t, A_t, \boldsymbol{w})^\top \boldsymbol{w} - \hat{y})^2$
14:     $loss_w = (\phi^\top \boldsymbol{w} - R_{t+1})^2$
15:     Gradient descent on $\psi_\theta$ and $\boldsymbol{w}$
16:     Set $\psi_{\theta_{u_1}} \leftarrow \psi_\theta$
17:     Update the parameters of the consolidation parameters analytically using Eq.11, Eq. 12, Eq.13 (Stochastic Gradient Descent)
18:     Set $\psi_\theta \leftarrow \psi_{\theta_{u_1}}$
19: **end for**

---

# D ENVIRONMENTS

## D.1 SLIPPERY 3D FOUR ROOMS ENVIRONMENT

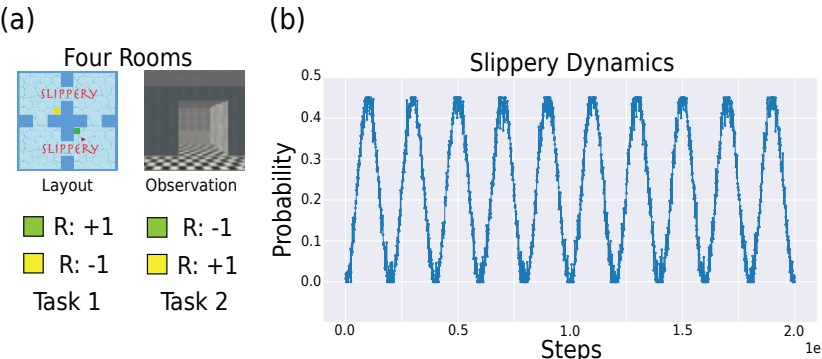

Figure 9: **(a)** The slippery variant of the 3D Four Rooms environment. The agent alternates between two tasks: in Task 1, reaching the green box produces +1 reward and the yellow box -1, in Task 2, the reward assignment is reversed. At each step, the agent's chosen action may be randomly replaced with a probability sampled from the noisy sine function shown in B. The agent receives only egocentric pixel observations. **(b)** A noisy sine wave that generates continuously varying slip probabilities, used to stochastically replace the agent's intended actions.

## D.2 CONTINUOUS CONTROL IN MUJOCO

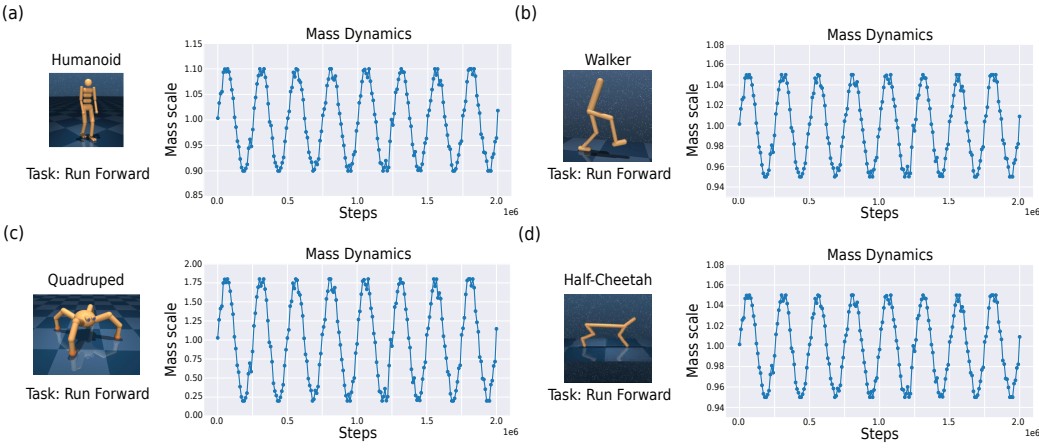

Figure 10: MuJoCo suite with continuous mass changes during training and evaluation for (a) Humanoid, (b)Walker, (c) Quadruped, (d) Half-cheetah. A noisy sine wave that generates continuously varying mass values, used to stochastically scale the agent's mass during training and evaluation.

# E EXPERIMENTAL DETAILS

In this section, we provide more details about the environments used in our experiments.

## E.1 3D SLIPPERY FOUR ROOMS ENVIRONMENT

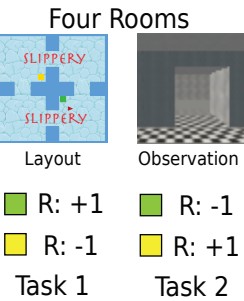

Figure 11: Slippery Four Rooms environment

We extended this environment used in the Simple Successor Features (Chua et al., 2024) which was built upon the original 3D Miniworld environment (Chevalier-Boisvert et al., 2023). In the slippery variant of the Four Rooms environment[1], we mimic wet or icy conditions in all four rooms, rather than just two rooms (top right and bottom left). The key difference is, unlike in the prior setup which uses a constant pre-determined slippery probability value, we sample the slippery probabilities using a noisy sine function to ensure continuous changes during training and evaluation. We kept the same two tasks structure, where the rewards alternate when the task switches. The agent receives egocentric pixel observations at every step and the actions are moving forward, backwards, turning left and turning right.

The specific parameters defining the 3D Slippery Four Rooms Environment are detailed in Table 1.

Table 1: 3D Slippery Four Rooms Environment Specific Parameters

| PARAMETER | VALUE |
| --- | --- |
| OBSERVATION TYPE | EGOCENTRIC |
| FRAME STACKING | NO |
| RGB OR GREYSCALING | RGB |
| NUM TRAINING FRAMES PER TASK | 5 MILLION FRAMES |
| NUM EXPOSURE | 2 |
| NUM TASK PER EXPOSURE | 2 |
| BATCH SIZE | 32 |
| $\epsilon$ DECAY | 1 MILLION FRAMES |
| ACTION REPEAT | NO |
| ACTION DIMENSION | 4 |
| OBSERVATION SIZE | $84 \times 84$ |
| MAX FRAMES PER EPISODE | 4000 |
| TASK LEARNING RATE | 0.001 |
| SLIPPERY PROBABILITY RANGE | 0 TO 0.45 |

## E.2 MUJOCO

In this work, we consider only state observations. For the embodiments, we chose both Walker, Half-Cheetah, Quadruped and Humanoid. We broadly follow the same setup as Yarats et al. (2021) and Chua et al. (2024), and include their models as baselines, which we denote as "DDPG" and "SFs" respectively.

---

[1]https://github.com/raymondchua/miniworld_four_rooms

The codebase was adapted from the Simple Successor Features repository[2]. The specific parameters we used in the Mujoco environment for training broadly follow the ones defined in Chua et al. (2024), with some exceptions, such as the training steps and the learning rate for the task encoding vector. The specific parameters for MuJoCo are detailed in Table 2.

Table 2: Mujoco Environment Specific Parameters

| PARAMETER | VALUE |
|---|---|
| FRAME STACKING | YES |
| OBSERVATION | STATE |
| NUM TRAINING STEPS PER TASK | 2 MILLION STEPS |
| NUM EXPOSURE | 1 |
| ACTION REPEAT | 1 |
| BATCH SIZE | 1024 |
| FEATURE DIM | 128 |
| HIDDEN DIM | 1024 |
| MAX STEPS PER EPISODE | 10000 |
| SF DIM | 10 |
| TASK LEARNING RATE | HALF-CHEETAH: 1E-3, WALKER: 1E-8, QUADRUPED: 1E-9, HUMANOID:1E-8 |
| TASK UPDATE FREQUENCY | 10 |

_______________

[2]https://github.com/raymondchua/simple_successor_features

# F PLASTICITY-STABILITY ANALYSIS IN NON-STATIONARY CONDITIONS

## F.1 SLIPPERY FOUR ROOMS ENVIRONMENT

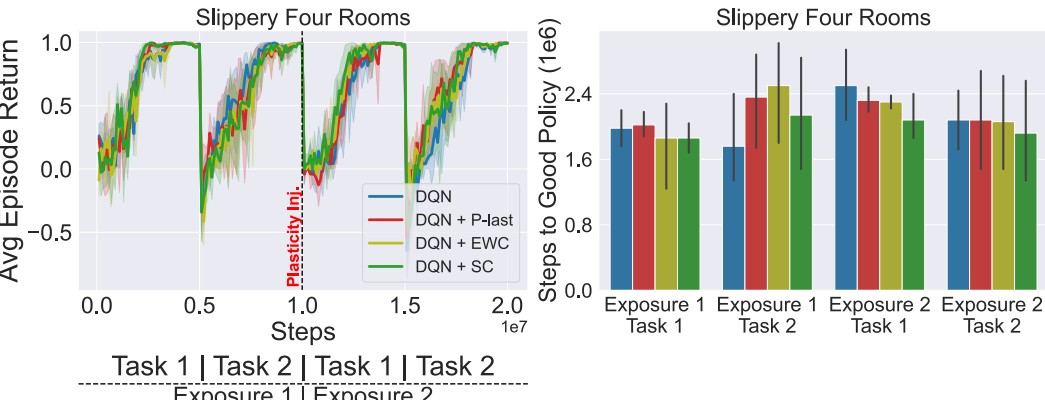

Figure 12: Plasticity–stability analysis in the Slippery Four Rooms environment. The DQN agent undergoes two exposures; after each learning phase, the reward mapping is reversed. **(a)** Average return per episode. **(b)** Learning efficiency (steps to reach a good policy; lower is better). For the plasticity-injection agent, plasticity was injected once at 10 million environment steps (end of Exposure 1). In both panels, the DQN agent with synaptic consolidation (SC) learned a good policy in fewer steps than the plasticity-injection agent in the second exposure.

## F.2 MuJoCo Suite Results

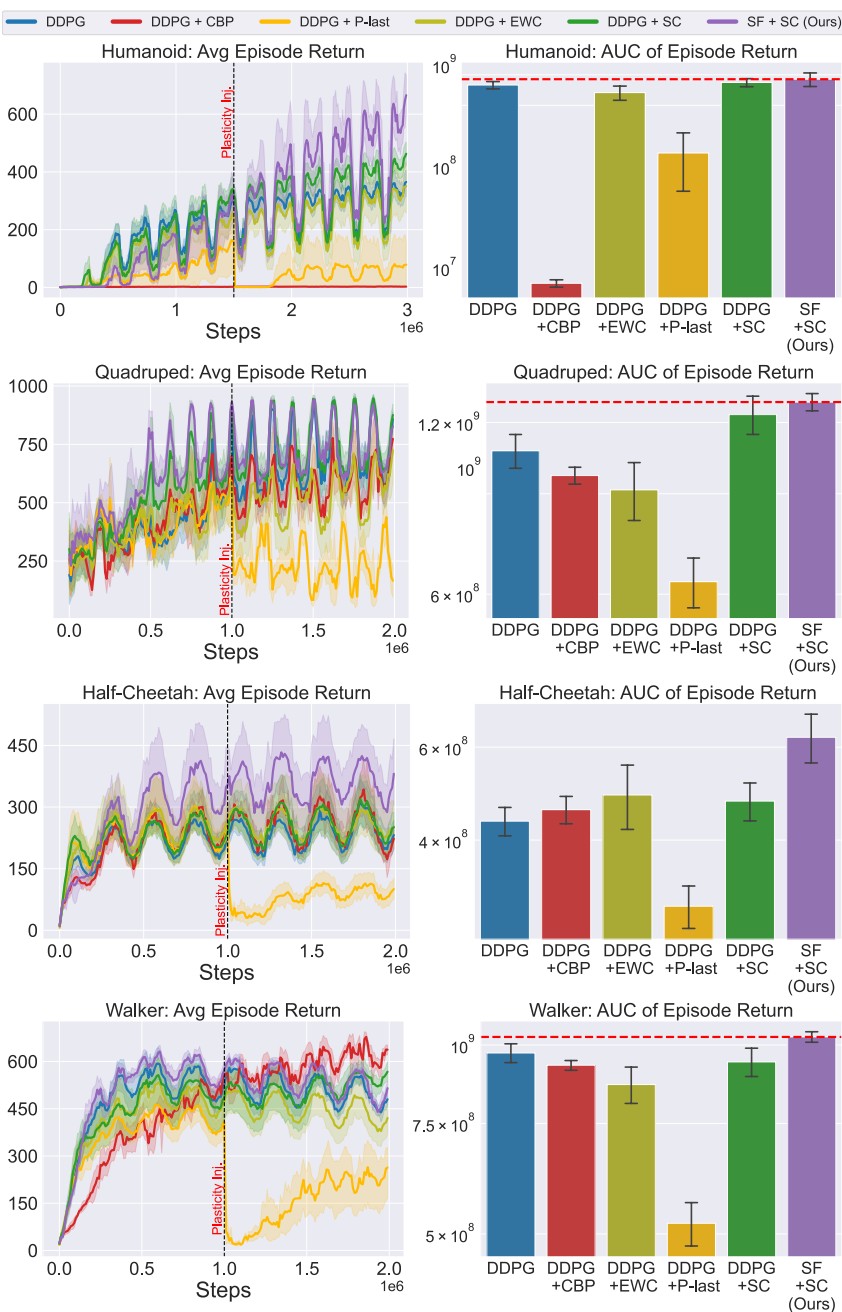

Figure 13: Plasticity–stability analysis in the MuJoCo suite under continuous mass changes during training and evaluation. We compare a baseline DDPG agent with three variants: (i) Continual Backprop (CBP), which selectively resets least-active weights; (ii) plasticity injection by resetting the weights in the last layer (P-last); and (iii) synaptic consolidation (SC). Plasticity injection was not evaluated in Humanoid and Quadruped, as it failed to yield meaningful learning in Walker and Half-Cheetah. Across tasks, CBP generally struggled to outperform DDPG, with complete collapse in Humanoid, while SC improved stability.

# G SCHEMATIC OF SYNAPTIC CONSOLIDATION FOR Q-VALUES VS. SUCCESSOR FEATURES

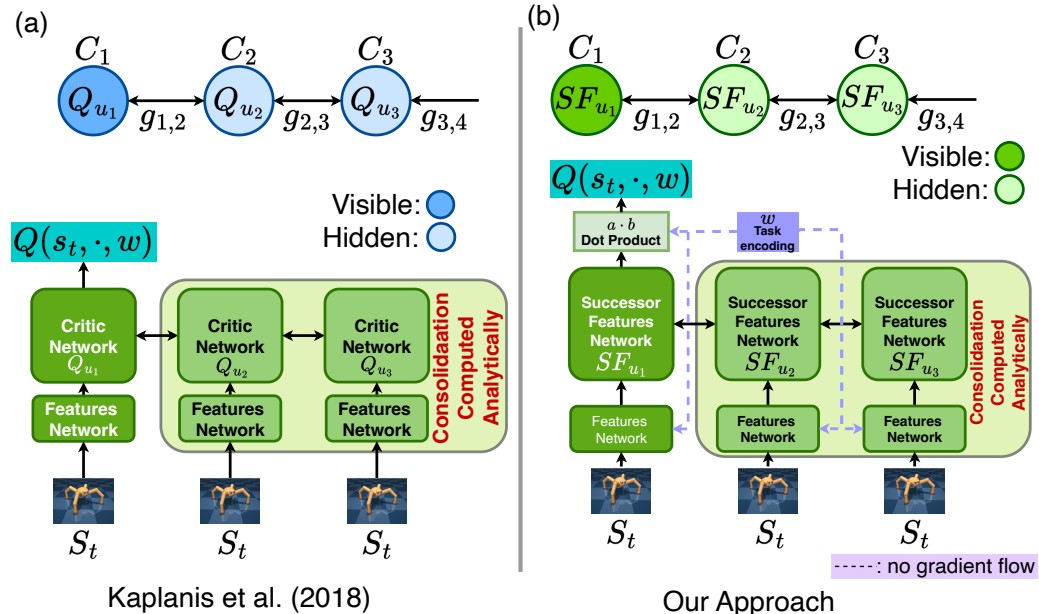

Figure 14: Schematic of synaptic consolidation applied to Q-values and to Successor Features (SFs). **(a):** (Kaplanis et al., 2018) showed that adapting the synaptic consolidation mechanism of (Benna & Fusi, 2016) to Q-values improves robustness in continual RL. **(b):** Here, we extend this approach to predictive, generalizable representations using Simple Successor Features (Chua et al., 2024). Consolidated variables (e.g., $Q_{u_2}, Q_{u_3}, \ldots$ or $SF_{u_2}, SF_{u_3}, \ldots$) are computed analytically and therefore lie outside the computational graph used to update $Q_{u_1}$ or $SF_{u_1}$ by backpropagation.

## H   Q-VALUES VS. SFS, ELASTIC WEIGHT CONSOLIDATION VS SYNAPTIC CONSOLIDATION COMPARISON

### H.1   SLIPPERY FOUR ROOMS ENVIRONMENT

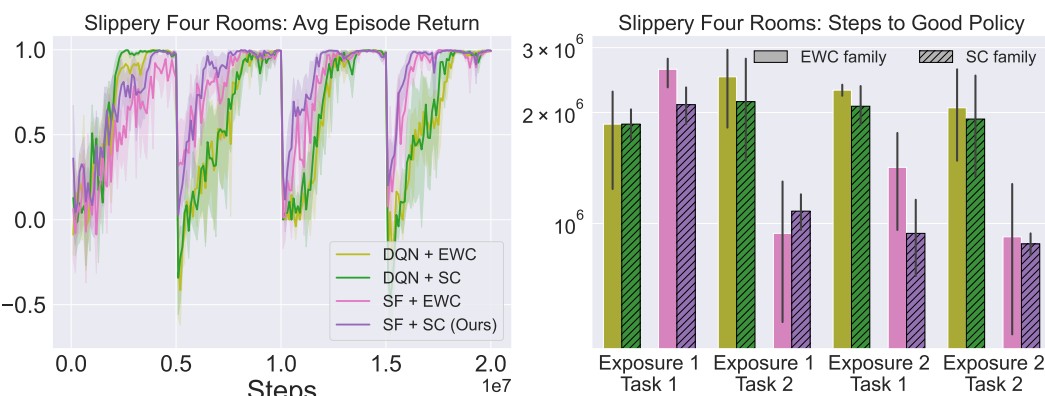

Figure 15: Comparison of Q-values and Successor Features (SFs), with synaptic consolidation (SC) or elastic weight consolidation (EWC), in the 3D Slippery Four Rooms environment during training and evaluation. Applying SC to Q-values (green) and SFs (purple) offers higher learning efficiency than their EWC counterparts, requiring fewer steps to learn a good policy. This demonstrates that SC is more effective than EWC, particularly when applied to SFs, during the agent's second exposure to the tasks.

H.2 MUJOCO SUITE WITH CONTINUOUS MASS CHANGES

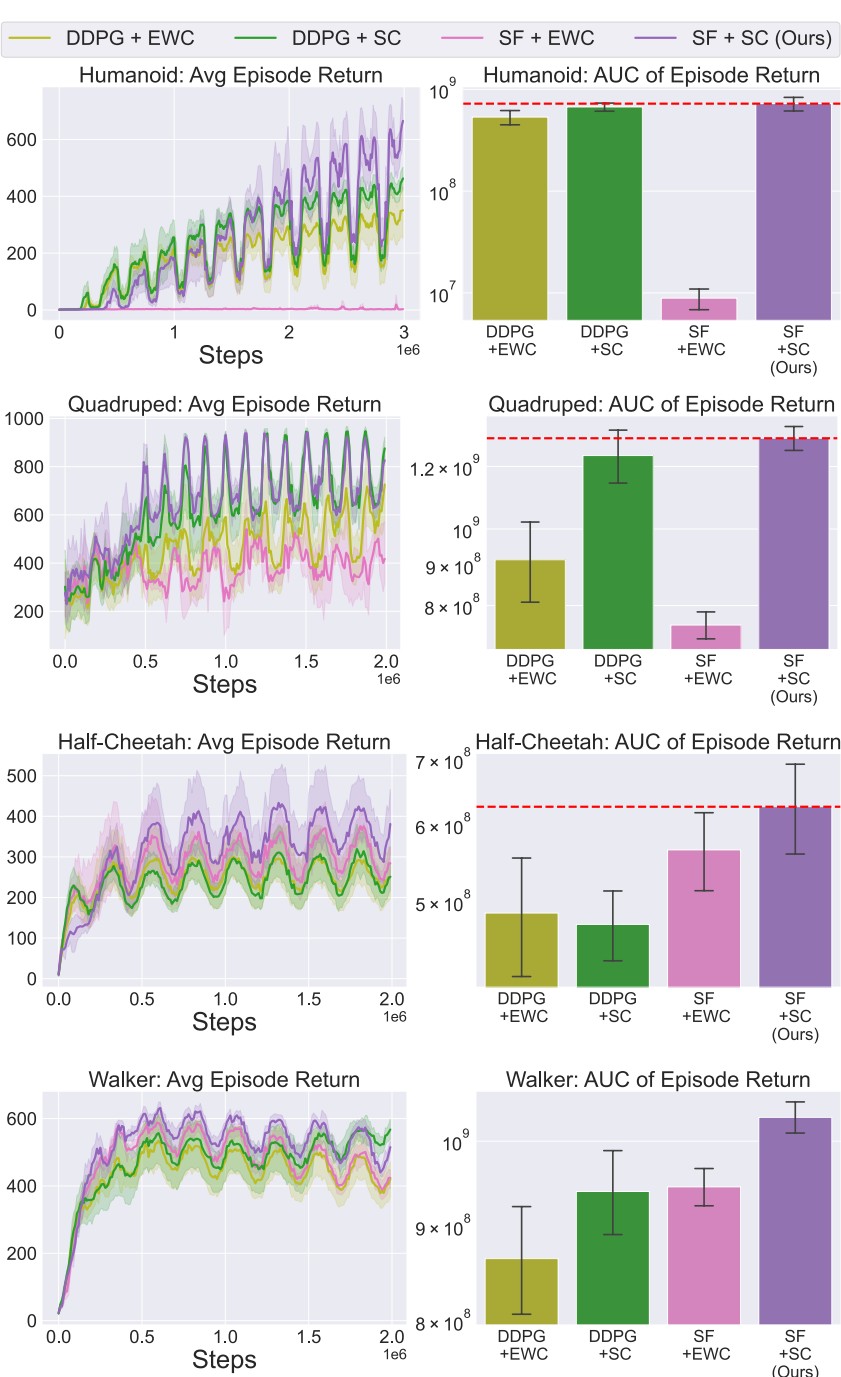

Figure 16: Comparison of Q-values and Successor Features (SFs), with and without synaptic consolidation, on the MuJoCo suite under continuous mass changes during training and evaluation. Interestingly, unlike Q-values, applying synaptic consolidation to SFs (purple) yields consistently higher learning efficiency.

# I Q-VALUES VS. SFS, WITH AND WITHOUT SYNAPTIC CONSOLIDATION COMPARISON

In this section, we present the results of apply synaptic consolidation to the parameters of Q-values versus the parameters of Successor Features.

## I.1 SLIPPERY FOUR ROOMS ENVIRONMENT

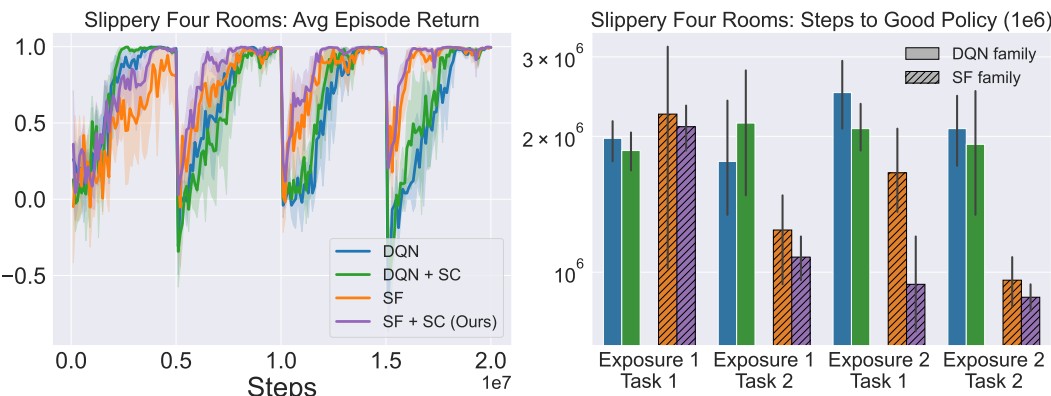

Figure 17: Comparison of consolidating the parameters of Q-values and SFs using Synaptic Consolidation (SC) using the 3D Slippery Four Rooms environment. **(left):** Average episode return plot. **(right):** Number of trainng steps needed to reach a pre-determined good policy. Lesser steps the better. Applying SC to the SFs (purple) yield better learning performance overall.

## I.2  MUJOCO SUITE WITH CONTINUOUS MASS CHANGES

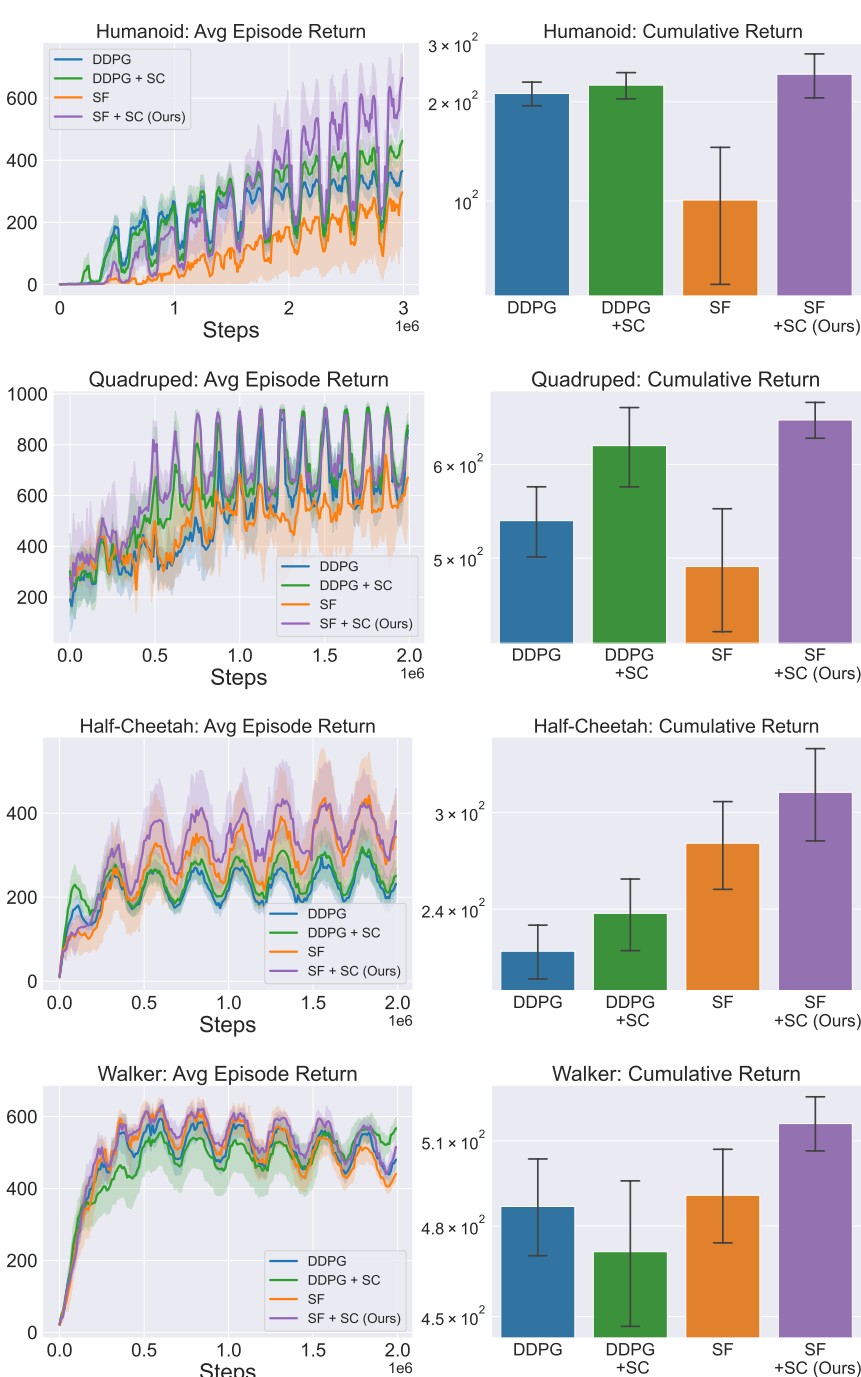

Figure 18: Comparison of consolidating the parameters of Q-values and SFs using Synaptic Consolidation using the MuJoCo suite. Interestingly, when compared to DDPG (blue), SFs (orange) learns well in Half-Cheetah and Walker but not Quadruped and Humanoid. This is probably due to higher complexity in Quadruped and Humanoid as they have larger state and action spaces. Overall, applying SC to the SFs (purple) yield better learning performance, highlighting their effectiveness when combined together.

## J ANALYSIS OF FAST AND SLOW TIMESCALE VARIABLES

### J.1 SLIPPERY FOUR ROOMS ENVIRONMENT RESULTS

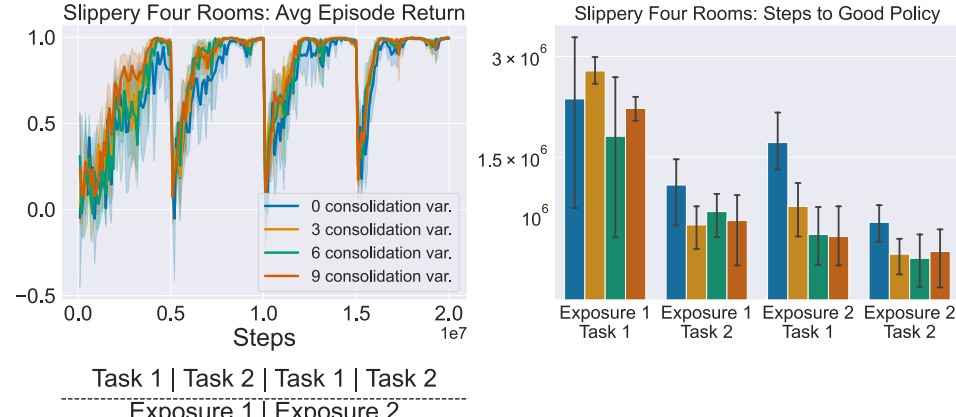

Figure 19: Analysis of fast and slow timescale variables in the 3D Slippery Four Rooms environment during training and evaluation. Using synaptic consolidation clearly leads to better learning efficiency, but there is no clear advantage between six and nine consolidation variables.

## J.2 MUJOCO SUITE RESULTS

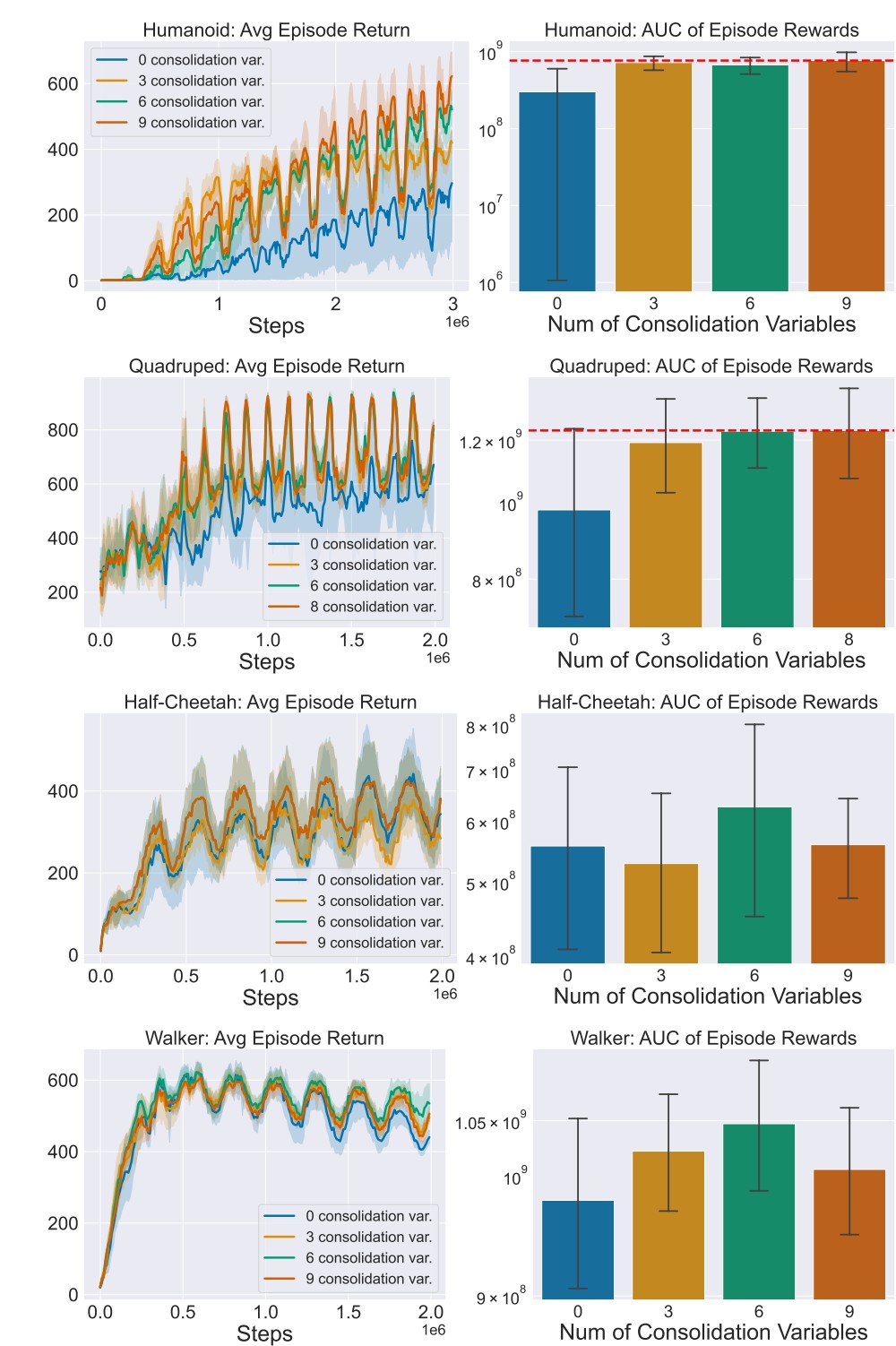

Figure 20: Analysis of fast and slow timescale variables on the MuJoCo suite under continuous mass changes during training and evaluation. Using more consolidation variables (six, eight or nine) yields consistently higher learning efficiency, highlighting the importance of slower-timescale variables.

# K ARCHITECTURE FOR RECALLING CONSOLIDATED SUCCESSOR FEATURES USING CROSS-ATTENTION

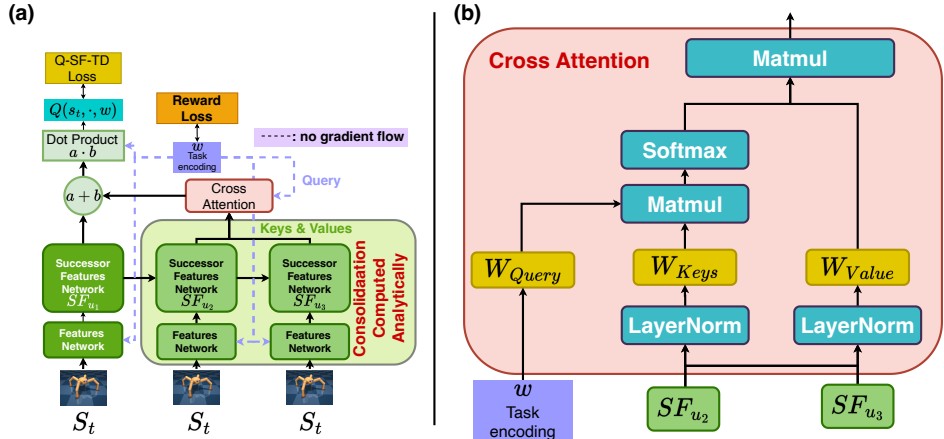

Figure 21: Using cross-attention to recall information from the SF consolidation modules. (**a:** A high-level schematic on how the cross-attention mechanism is used. (**b:** The computations for the cross-attention mechanism. We used the task encoding vector $w$ as the query, the SFs consolidation variables except the most plastic one as keys and values ($SF_{u_2}, SF_{u_3}, \ldots, SF_{u_K}$). Because these SFs consolidation variables are computed analytically, they are not part of the computational graph. Therefore, we apply a reparameterization trick to add the output of the cross-attention mechanism to the most plastic SF ($SF_{u_1}$) so that the learnable weights ($W_{Query}, W_{Keys}, W_{Values}$) in the cross-attention mechanism are learned via the Q-SF-TD loss (Eq. 5).

## L    CROSS-ATTENTION ANALYSIS OF FAST AND SLOW TIMESCALE VARIABLES

### L.1    SLIPPERY FOUR ROOMS ENVIRONMENT RESULTS

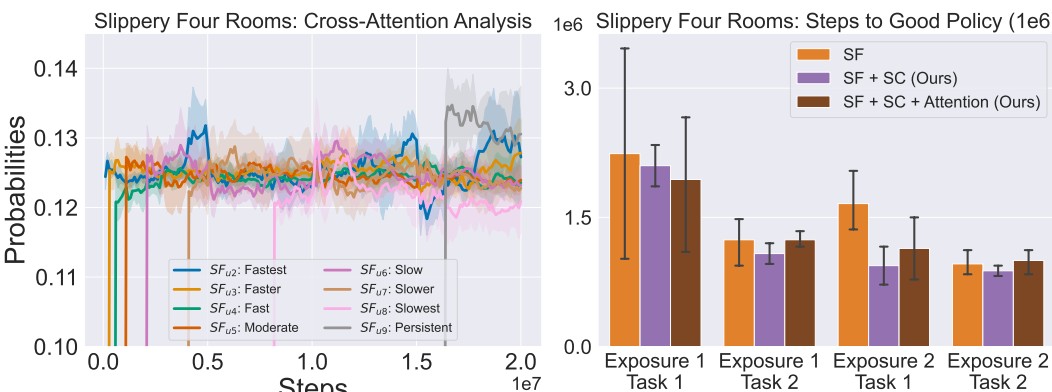

Figure 22: Analysis of all consolidated variables using Cross-Attention during training in the 3D Slippery Four Rooms environment. The cross-attention probabilities indicate that fast and slow timescale variables were attended to similarly, suggesting nearly equal contribution. This may be due to the sparse reward structure in the 3D Slippery Four Rooms environment, which affects how discriminate the SFs are given that the SFs are learned via the reward signal using Q-SF-TD loss (Eq. 5)

## L.2 MUJOCO SUITE RESULTS

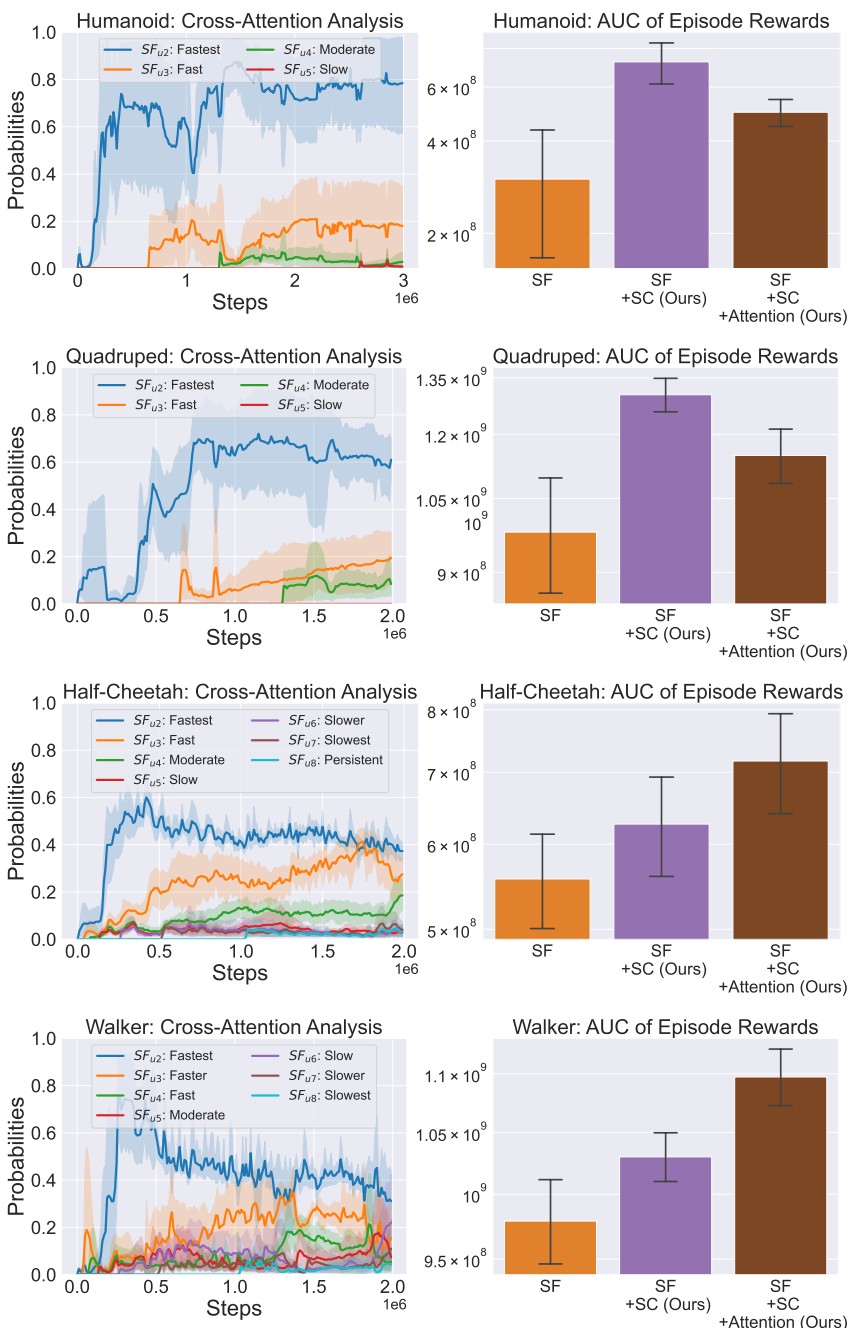

Figure 23: Analysis of all consolidated variables using cross-attention in the MuJoCo suite under continuous mass changes. Memory recall was performed solely through the cross-attention mechanism, rather than by waiting for information to propagate from slower to faster timescale variables. Unsurprisingly, faster timescale variables were attended to more than slower ones. Notably, Half-Cheetah and Walker benefited from memory recall via cross-attention, whereas Quadruped and Humanoid require more steps to learn. This may be due to the higher complexity of Quadruped and Humanoid, as their larger state and action spaces reduce the learning efficiency of the cross-attention mechanism.

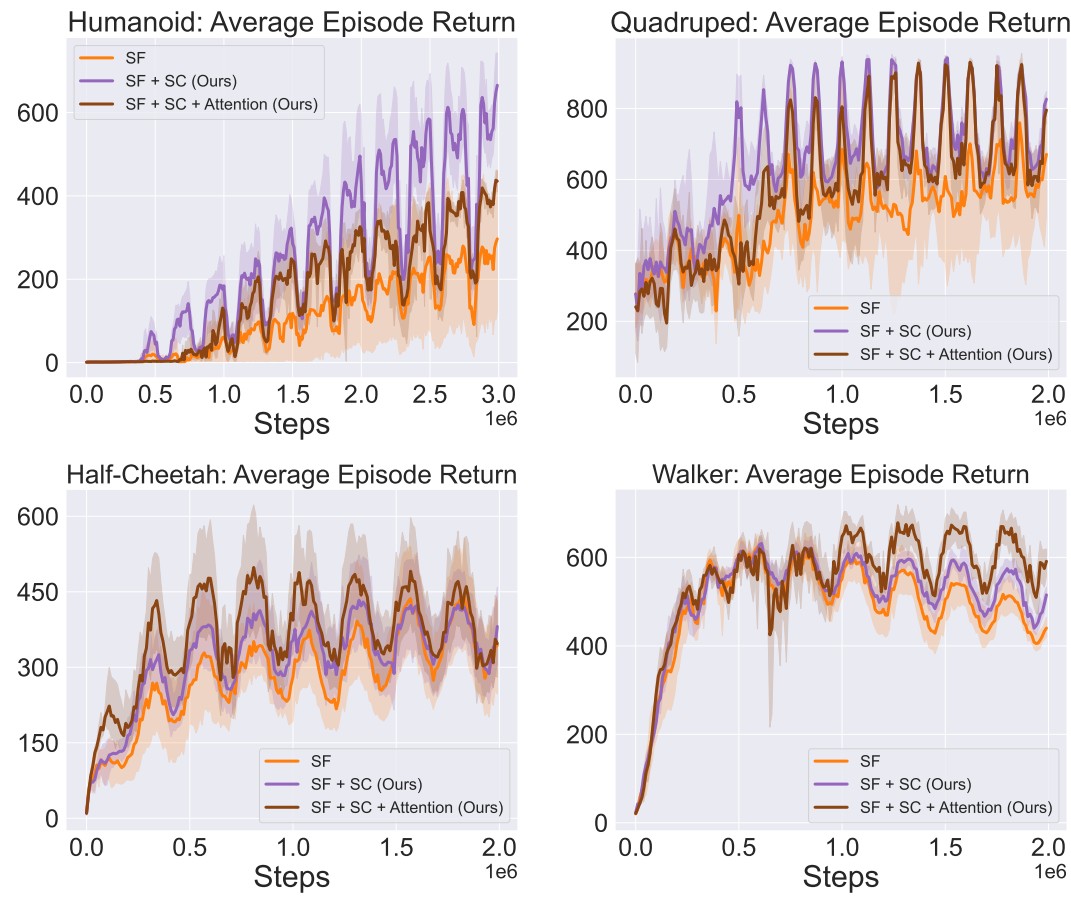

Figure 24: Learning curves in the MuJoCo suite under continuous mass changes with cross-attention over consolidated variables. Faster timescale variables were generally attended to more strongly than slower ones as shown in Figure 23. Half-Cheetah and Walker benefited from cross-attention–based memory recall, whereas Quadruped and Humanoid require more steps to learn, likely due to their higher complexity and larger state–action spaces which reduces the learning efficiency of the cross-attention mechanism.

# M  AGENTS

In this section, we describe our agent as well as the ones we used for comparisons.

## M.1  SUCCESSOR FEATURES WITH SYNAPTIC CONSOLIDATION

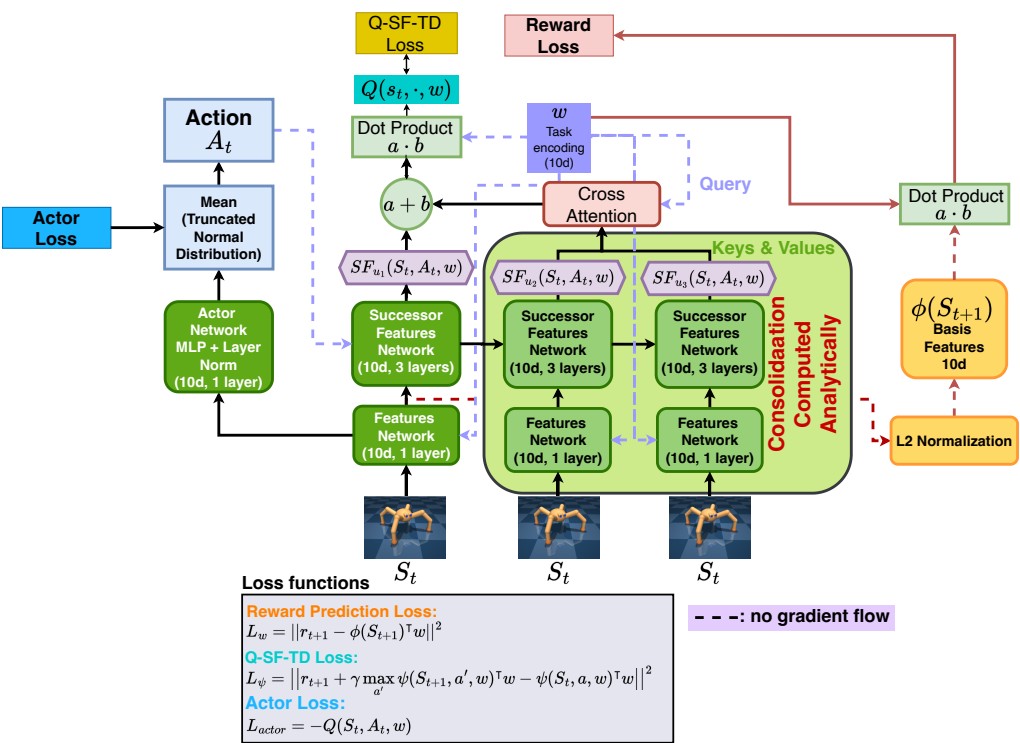

Figure 25: Simple SFs with synaptic consolidation architecture. Simple SFs were adapted from (Chua et al., 2024), with DDPG (Lillicrap et al., 2015) as base model. The synaptic consolidation variables are updated analytically (see section 4 for more details on the consolidation variables).

We swept the task learning rate of the task encoding vector across the values of $\{10^{-5}, 10^{-6}, \ldots, 10^{-10}\}$ when optimizing the reward prediction loss (Eq. 4) for the MuJoCo suite. In the naturalistic, continual non-stationary setting that we are studying, we find that a lower learning rate for learning the task encoding vector $w$ generally helps.

The DQN variant largely follows the same architecture, but without the actor network. The only major difference is that hidden dimensions are set to 256. The encoder architecture is the same as DQN encoder.

Table 3: Simple SF with Synaptic Consolidation Hyperparameters

| PARAMETER | VALUE |
|---|---|
| OPTIMIZER | ADAM (KINGMA & BA, 2014) |
| DISCOUNT($\gamma$) | 0.99 |
| REPLAY BUFFER SIZE | 1 MILLION |
| DOUBLE Q | YES (FUJIMOTO ET AL., 2018) |
| TARGET NETWORK: UPDATE PERIOD | 1000 |
| TARGET SMOOTHING COEFFICIENT | 0.01 |
| MIN REPLAY SIZE FOR SAMPLING | 5000 |
| LEARNING RATE | 0.0001 |
| NUMBER OF CONSOLIDATION VARIABLES | 9 |
| FLOW STRENGTH ($g_{1,2}$) | 0.125 |
| CAPACITY FOR THE FIRST VARIABLE ($C_1$) | 2 |
| LEARNING RATE FOR CONSOLIDATION | 1 |
| CAPACITY SCALING FACTOR ($s$) | 20 |
| BASIS FEATURES $\phi$ | L2-NORMALIZE (OUTPUT OF ENCODER) |
| FEATURE $\phi$ DIMENSION | 10 |
| FEATURES-TASK NETWORK HIDDEN UNITS | $\{1024, 10\}$ |
| FEATURES-TASK NETWORK NORMALIZATION | LAYER-NORM |
| FEATURES-TASK NETWORK NON-LINEARITY | TANH |
| SF $\psi$ DIMENSION | 10 |
| SF $\psi$ NETWORK HIDDEN UNITS | $\{1024, 1024, \text{SF } \psi \text{ DIM}\}$ |
| SF $\psi$ NETWORK NON-LINEARITY | RELU |

Table 4: Task $w$ encoding Hyperparameters

| PARAMETER | VALUE |
|---|---|
| TASK $w$ DIMENSION | 10 |
| TASK $w$ LEARNING RATE | ENVIRONMENT-DEPENDENT (SEE TABLE 1 & 2) |
| TASK $w$ OPTIMIZER | ADAM (KINGMA & BA, 2014) |

## M.2 ELASTIC WEIGHT CONSOLIDATION

For the Elastic Weight Consolidation (EWC) agents (Kirkpatrick et al., 2017), we adapt the online variant (Schwarz et al., 2018), such that instead the fisher information is computed at every $k$ steps instead of waiting till the end of the task, thus removing the need for tasks boundaries information. The EWC loss is defined as:

$$L(\theta) = L_{TD}(\theta) + \sum_i \frac{\lambda}{2} F_i (\theta_i - \theta_i^*)^2 \tag{31}$$

where $i$ is the number of parameters, $\lambda \in \mathbb{R}$ is the regularization factor, $L_{TD}$ can be either the DQN loss or Q-ST-TD loss (Eq. 5), if we are learning SFs. The fisher information is computed by squaring the gradients of the parameters. In our experiments, we set the fisher computation interval to every 10k steps, which is the number of steps per episode in MuJoCo. We also swept the regularization factor $\lambda \in \{12, 25, 75, 125, 175\}$, following the same set up as (Schwarz et al., 2018). In our experiments, we find $\lambda = 25$ works best.

## M.3 PLASTICITY INJECTION MODEL

For the plasticity injection models, we consider plasticity injection on the last layer (P-last) for both the Slippery Four Rooms environment and the MuJoCo suite. We follow the same setup as (Nikishin et al., 2023), whereby during the plasticity injection step $k$, we retain freeze the parameters of last layer in the artificial neural network $\theta$ and introduce a new set of parameters $\theta'$ which is sampled from random initialization. The set of new of parameters is then further copied, such that we will

have $\theta' = \theta'_1 = \theta'_2$ and the output of the network is computed using:

$$h_\theta(x) + h_{\theta'_1}(x) - h_{\theta'_2} \tag{32}$$

where $h$ is the function of the artificial neural network and $x$ is the input. After the plasticity injection step $k+1$ and beyond, only $\theta'_1$ is allowed to be updated while $\theta', \theta'_2$ are kept frozen, leading $(\theta' - \theta'_2)$ to be the bias term.

We experimented with injecting plasticity at 25%, 50% and 75% of the training steps, but observed little effect in our setting. We also found that this method was ineffective when adapted to DDPG and evaluated in MuJoCo. Therefore, we included Continual Backprop (Dohare et al., 2024) as additional baseline for MuJoCo.

## M.4 CONTINUAL BACKPROP

The plasticity injection method described above can hurt learning performance when injecting plasticity into the critic network of actor-critic architecture. Therefore, we consider another baseline model, known as continual backprop (CBP), that is more effective in enhancing plasticity (Dohare et al., 2024). Rather than re-initializing parameters at random, CBP selectively injects plasticity into parameters who were least useful for the current task. This least useful metric is known as the contribution utility, which measures both the hidden unit's activity and its outgoing connection strength. For each hidden unit $i$ in layer $l$ at time $t$, the contribution utility is defined as:

$$u_l[i] = \eta \times u_l[i] + (1 - \eta) \times |h_{l,i,t}| \times \sum_{k=1}^{n_{l+2}} |w_{l,i,k,t}| \tag{33}$$

where $h_{l,i,t}$ is the output of the $i$th hidden unit in layer $l$ at time $t$, $w_{l,i,k,t}$ is the weight connecting the $i$th unit in layer $l$ to the $k$th unit in layer $l + 1$ at time $t$ and $n_{l+1}$ is the number of units in the next layer $l + 1$. This contribution utility can be thought of as a running average of instantaneous contributions, with a decay rate $\eta$.

At each step, CBP identifies eligible units for re-initialization based on two criteria, first is the low contribution utility which indicates that the unit has not been useful in recent training phase, and second is the lifespan, which ensures that units are only re-initialized after allowing to have a sufficient steps of learning. By periodically resetting under-utilized units, CBP ensures that the network maintains plasticity throughout learning. In our experiments, we broadly follow the parameters defined in (Dohare et al., 2024). We swept the replacement rate across the values of $\{10^{-3}, 10^{-4}, 10^{-5}\}$ but observed little effect in our setting, so we kept it at $10^{-4}$. We also swept the maturity threshold across the values of $\{100, 1000, 10000\}$ and also observed little effect, so we kept it as 1000.

In our experiments, unsurprisingly, we found CBP to be much more effective than the plasticity injection model (P-last) introduced in section M.3. Particularly in Quadruped, Half-Cheetah and Walker (Figure 13), we see that CBP outperforms P-last. However, across MuJoCo tasks, CBP generally failed to outperform its base model, DDPG, suggesting that in our natural and continually evolving environments, stability rather than plasticity, is the critical bottleneck, since injecting plasticity did not improve performance.

## N  IMPLEMENTATION DETAILS

For our experimental setup, we used Python 3 (Van Rossum & Drake, 2009) as the primary programming language. The agents and the training framework were developed using Jax (Bradbury et al., 2018; Godwin* et al., 2020) for the Slippery Four Rooms environment and PyTorch (Paszke et al., 2019) for the MuJoCo embodiments (Todorov et al., 2012). We used the DeepMind Control Suite (DMC) (Tunyasuvunakool et al., 2020) to manipulate the embodiments. Flax (Heek et al., 2024) was employed for implementing the neural network components. For data visualization, we used Seaborn (Waskom, 2021), Matplotlib (Hunter, 2007) on Jupyter Notebooks (Kluyver et al., 2016) to generate the plots. The configuration and management of our experiments were performed with Hydra (Yadan, 2019) and Weights & Biases (Biewald, 2020) respectively. All experiments were conducted using Nvidia A100 GPUs and completed within one to two days max. The code used in the study will be released in the near future, following an internal review process.

# O  COMPUTATIONAL COMPLEXITY

In this section, we report the computational complexity of all methods, measured in frames per second (FPS) during training. Across both the Slippery Four Rooms and the Mujoco Humanoid experiments, our model (SF + SC) exhibits the lowest throughput. This slowdown arises from the consolidation dynamics, which require analytical, sequential updates between consolidation variables, in addition to learning the SFs and the task-encoding vector $w$. Because these updates are not handled by backpropagation and cannot be parallelised, they introduce a constant-factor computational overhead inherent to the mechanism. Moreover, this overhead increases with the number of consolidation variables, where using more variables leads to proportionally lower FPS.

## O.1  SLIPPERY FOUR ROOMS ENVIRONMENT

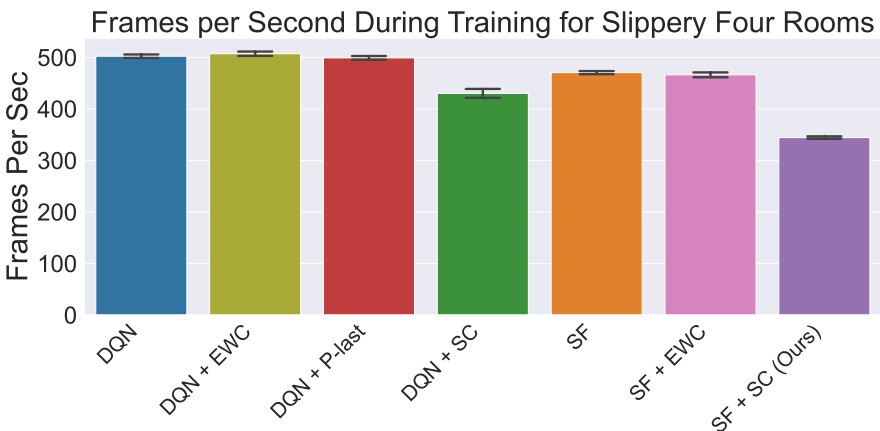

Figure 26: Comparison of training throughput (FPS) for all models in the Slippery Four Rooms environment. Higher FPS reflects more efficient computation.

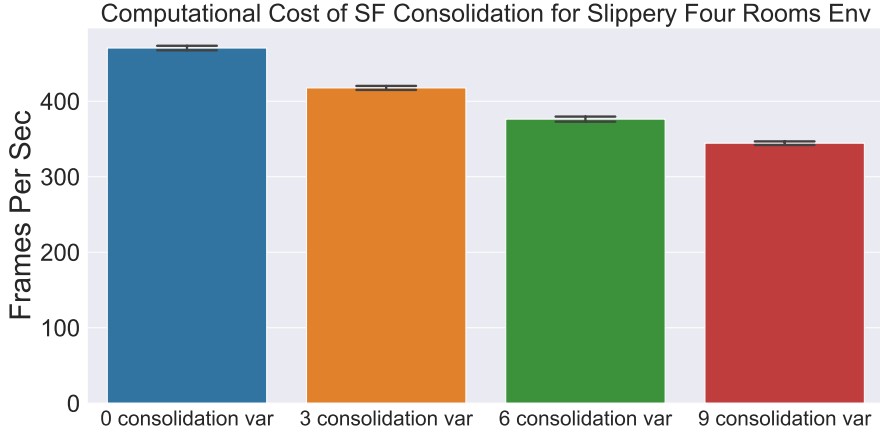

Figure 27: Comparison of training throughput (FPS) for different number of consolidation variables for the SFs within the slippery four rooms environment. Higher FPS reflects more efficient computation.

## O.2  MUJOCO - HUMANOID

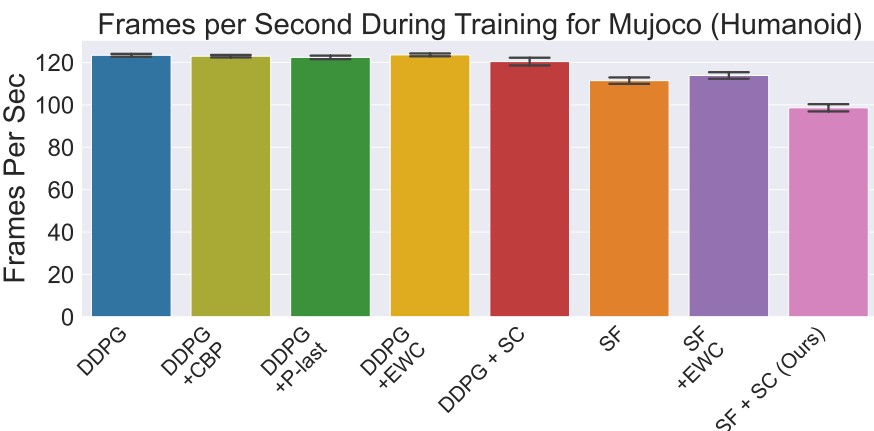

Figure 28: Comparison of training throughput (FPS) for all models in the humanoid embodiment within Mujoco environment. Higher FPS reflects more efficient computation.

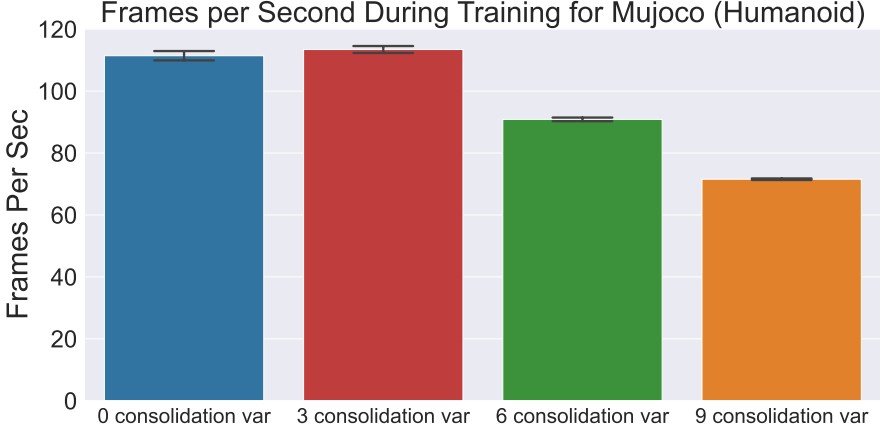

Figure 29: Comparison of training throughput (FPS) for different number of consolidation variables for the SFs within the humanoid embodiment within Mujoco environment. Higher FPS reflects more efficient computation.

