# OpenReview forum: "Balancing Plasticity and Stability with Fast and Slow Successor Features"
_ICLR.cc/2026/Conference — Submitted to ICLR 2026_

### Official Review · Reviewer_P8uF · 2025-10-29

**Soundness:** 2
**Presentation:** 3
**Contribution:** 3
**Rating:** 4
**Confidence:** 3

**Summary:**

This paper proposes the use of successor features (SF) along with synaptic consolidation(SC) for the stability-plasticity tradeoff. It shows that consolidation is more effective when done on parameters of SF rather than Q-values, in DQN and DDPG.
It also compares their method (SF+SC) with baselines that either generally help with stability (e.g., elastic weight consolidation or synaptic consolidation) or plasticity (continual backprop or plasticity injection) and shows that their method outperforms in most cases.
It also motivates the use of environments where change happens gradually over time, not abruptly, and uses those settings throughout the experiments.

**Strengths:**

1. The motivation to use environments that reflect the natural, gradual change, as they occur in real-world settings, is valuable.

2. The paper is well-written,  covers the related literature, and the idea to apply consolidation to SF instead of Q-values is relatively simple and potentially helpful.

**Weaknesses:**

1. The paper emphasizes the importance of stability in the experiments; however, stability and plasticity are not decoupled and evaluated quantitatively in the paper. As mentioned in the paper, some methods help more with plasticity, and some more with stability. However, it is valid but incomplete to determine which one performs better in an experiment and conclude that the underlying reason for failure is what the method is designed to address. It would help to define some metrics for plasticity (e.g., how fast the agent adapts after a change in the environment) and stability (e.g., forgetting, or performance drop in a familiar previous setting after being trained on new data), to support the idea that stability is playing a more crucial role.

2. Environments without task boundaries are motivated in the paper, but in the four rooms experiments, two tasks are defined on top of the gradual non-stationarity, and the agent goes through a multi-task environment. Also, most of the agents reach a good performance in each task at the end of the task duration, which suggests that the non-stationarity generated by the slippery actions does not create a dynamic complex enough to break the agents' performance. How can you modify this environment such that it’s still a challenging and suitable testbed for gradual non-stationary, and without the notion of tasks?

3. The paper would benefit from statistical tests, e.g., for the bar plots, especially since the number of seeds is limited.

Minor comments:
1. Continuous may not be a good word for referring to the types of environments aimed at in this paper. continual, natural non-stationary, or gradual might be better, as continuous environments might be mistaken with environments with continuous action spaces as opposed to discrete.

2. It would be helpful to add a comparison of the computational complexity of the methods.

3. Please cite some of the previous work that proposes or uses environments with gradual change, and without task boundaries, for continual/lifelong reinforcement learning.

**Questions:**

1. What metrics could be used to quantitatively measure plasticity and stability, beyond learning curves or AUC of episodic returns?

2. Why is CBP missing from the baselines in the Four Rooms experiment?

3. Some mitigation methods perform worse than the vanilla DQN or DDPG baselines (e.g., DDPG + P-last in the mujoco environments). Could you clarify why this might happen, and whether it is a tuning problem or is inherent to the algorithm?

4. In Figure 5, the legend lists six agents, but subfigure (a) appears to include seven curves. What does the pink line show?

---

> ### Author Response · Authors · 2025-11-29
>
> Thank you for taking the time to review our rebuttal. We sincerely appreciate your thoughtful comments and are glad to have the opportunity to provide further clarifications. Below, we respond to your excellent points. Please don’t hesitate to reach out if you have any additional questions or concerns.
>
> # 1. Continuous non-stationary environments may not be the best way
> We thank the reviewer for pointing out the potential ambiguity in our use of the term continuous environments, which may be interpreted as referring to continuous action spaces. To avoid confusion, we will replace “continuous” with “naturalistic continual” and add a clarification sentence specifying that we refer to temporal drift in dynamics, and not continuous control settings.

---

> > ### Author Response · Authors · 2025-12-04
> >
> > We have made the requested modifications in the revised manuscript. All the changes are in red text.

---

> ### Author Response · Authors · 2025-11-29
>
> # 2. Computational complexity of the methods
> We thank the reviewer for this helpful suggestion. We agree that reporting the computational complexity of the proposed methods would strengthen the paper. Providing an accurate analysis requires some additional work on our side, and we will include a complexity breakdown in the revised manuscript in the coming days and provide an update here.

---

> > ### Author Response · Authors · 2025-12-04
> >
> > We have now added the computational cost analysis to Section O in the appendix. We evaluated training throughput (FPS) in both the Slippery Four Rooms environment and the Humanoid embodiment in MuJoCo. Overall, we find that the consolidation-augmented model (SF + SC) is slower than the non-consolidated baseline, 26.79% slower than SF in Slippery Four Rooms and 11.50% slower in Humanoid.
> >
> > This slowdown is expected because the consolidation dynamics perform analytic, non-parallelisable updates across the consolidation variables at every training step. These updates are not handled by backpropagation and must be executed sequentially, introducing a constant-factor overhead inherent to the mechanism rather than an implementation inefficiency.
> >
> > Importantly, the overhead remains modest, does not increase with task complexity, and, as anticipated, increases predictably with the number of consolidation variables used. Despite the additional computation, learning is not hindered in any experiment, and the substantial gains in stability and reduced forgetting outweigh this cost.

---

> ### Author Response · Authors · 2025-11-29
>
> # 3. Citation of previous work that uses environments with gradual change
> Thank you for the suggestion. We will add the following citation which uses environments with gradual change to our manuscript. It is interesting to note that the gradual changes in this reference are also simulated using a sine function.
>
> [1] Xie, Annie, James Harrison, and Chelsea Finn. "Deep reinforcement learning amidst lifelong non-stationarity." arXiv preprint arXiv:2006.10701 (2020).

---

> ### Author Response · Authors · 2025-11-29
>
> # 4. Metrics to quantitatively measure plasticity and stability
> We thank the reviewer for this valuable suggestion.
>
> Although our Mujoco setting involves a single reward function, the continuously drifting mass parameter induces a sequence of quasi-stationary dynamics regimes, which enables us to meaningfully define stability and plasticity without requiring discrete task boundaries. Based on the reviewer’s comments, we plan to quantify plasticity as the within-regime adaptation gain, and stability as forgetting measured across sine cycles, defined as the performance drop when the agent encounters the same phase of the (noisy) sinusoidal mass drift in later cycles compared to its first occurrence.
> We are currently in the midst of running these analyses and will include the corresponding metrics and results in the revised manuscript once they are ready. We also would like to note that stability-preserving methods such as synaptic consolidation may still perform well on plasticity metrics but this does not imply that they are plasticity-enhancing methods. Rather, improved stability reduces interference and preserves relevant predictive structure, allowing the agent to start each new regime closer to an appropriate solution and thus adapt effectively without requiring many parameter changes.

---

> > ### Author Response · Authors · 2025-12-04
> >
> > We thank the reviewer for raising this important point.
> >
> > Due to unexpected technical issues with our compute cluster over the past two weeks, we were unfortunately unable to complete the additional quantitative analyses in time for the rebuttal.
> >
> > That said, we would like to clarify why defining stability and plasticity metrics in the MuJoCo benchmark is non-trivial and why forgetting cannot meaningfully be measured in this setting. The MuJoCo benchmark used in our paper is effectively a single-task continual learning scenario, where the underlying objective remains the same throughout training. Although the environment undergoes continuous drift in its dynamics, the task itself does not switch. As a result, forgetting (stability loss) cannot be observed in the usual sense, because there is no previously learned task that becomes unlearned. In this case, performance always reflects the current drifted dynamics.
> >
> > However, what can be measured is adaptation speed after changes in dynamics. All methods, including ours, exhibit some degree of adaptability (plasticity) under drift, although at different rates. Having said that, these adaptation rates alone do not allow us to disentangle whether a method’s reduced performance is due to insufficient plasticity or insufficient stability, because both factors interact under continuous drift.
> >
> > In particular, poor adaptation may arise either because the model fail to update effectively (plasticity issue) or because previous parameters over-constrain updates (stability issue). Thus, while adaptation curves could offer some insight, they would not constitute substantial evidence for attributing performance differences to stability vs. plasticity in this single-task drifting-dynamics setting.

---

> ### Author Response · Authors · 2025-11-29
>
> # 5. Continual Backprop for the four rooms environment
> We thank the reviewer for the helpful suggestion. We initially used plasticity injection (P-last) in the slippery four rooms environment because it was specifically designed for Deep Q-Network-style value based methods [2], which is the architecture that we used in this environment. In contrast, Continual Backprop (CBP) [3] was developed and evaluated primarily for actor-critic methods in reinforcement learning, and our experiments in MuJoCo show that P-last destabilizes DDPG while CBP remains stable. This suggests that CBP is better suited to continuous-control actor–critic models rather than DQN. For this reason, we paired each environment with the plasticity baseline most appropriate for its learning paradigm to ensure fair and meaningful comparisons.
>
> [2] Nikishin, Evgenii, et al. "Deep reinforcement learning with plasticity injection." Advances in Neural Information Processing Systems 36 (2023): 37142-37159.
>
> [3] Dohare, Shibhansh, et al. "Loss of plasticity in deep continual learning." Nature 632.8026 (2024): 768-774.

---

> ### Author Response · Authors · 2025-11-29
>
> # 6. Plasticity Injection methods (DDPG + P-last) perform worse than vanilla methods
> We thank the reviewer for raising this point. This relates to the above point, on why we didn’t include 5. Continual Backprop in the slippery four rooms environment.
>
> Plasticity injection was originally designed for value-based methods such as DQN [2], where resetting the last layer(s) restores plasticity without destabilizing the learning dynamics. In contrast, applying plasticity injection to actor–critic architectures like DDPG is much more disruptive because resetting the critic abruptly changes the value estimates that the actor depends on for policy updates. Even with a milder variant (resetting only the last critic layer), this still destabilized the actor and led to worse performance than the vanilla method (DDPG). This issue does not arise in DQN, where plasticity injection behaves as intended. However, to ensure a fairer comparison for actor-critic architectures, we included the continual backprop model (DDPG + CBP, red) for the MuJoCo benchmark, which periodically reset parameters for neurons that are less active. While this model (DDPG + CBP, red) performs better than DDPG + P-last (yellow), it still performs worse than DDPG (blue) in most non-humanoid environments (Figure 5) and fails to learn in the Humanoid environment (Figure 1).
>
> These results suggest that plasticity injection does not transfer well from value-based to actor–critic settings.
>
> [2] Nikishin, Evgenii, et al. "Deep reinforcement learning with plasticity injection." Advances in Neural Information Processing Systems 36 (2023): 37142-37159.

---

> ### Author Response · Authors · 2025-11-29
>
> # 7. Pink Curves in Figure 5
> We thank the reviewer for catching this issue. The pink curve corresponds to the SF + EWC model and was included in Figure 5 by mistake. Since we present the SF + EWC results separately in Figure 16 of Appendix H to avoid cluttering Figure 5, we will remove the pink curve from Figure 5 in the revised manuscript while keeping the correct results in Figure 16.

---

### Official Review · Reviewer_an5Z · 2025-10-29

**Soundness:** 2
**Presentation:** 3
**Contribution:** 2
**Rating:** 6
**Confidence:** 3

**Summary:**

This paper addresses continual reinforcement learning in environments with continuous non-stationarity, rather than the typical discrete task boundaries. The authors modify 3D Four Rooms and MuJoCo environments to incorporate smoothly changing dynamics using noisy sine functions. They investigate whether poor performance stems from loss of plasticity or stability, finding that stability-preserving methods outperform plasticity-focused ones. The main contribution is combining Successor Features (SFs) with a neuro-inspired synaptic consolidation mechanism across multiple timescales, which outperforms consolidating Q-values directly.

**Strengths:**

1. The paper tackles an important problem - real-world environments change gradually, not abruptly. The experimental design fairly compares stability vs plasticity mechanisms without requiring task boundary information, which is more realistic. The finding that stability matters more than plasticity in continuous settings contradicts recent emphasis on plasticity loss, providing valuable empirical insights.

2. The use of multi-timescale consolidation (not just dual fast/slow) is more biologically plausible and the cross-attention analysis provides useful interpretability about which timescales contribute to learning. The authors are honest about limitations - reporting when complex embodiments don't benefit from their approach adds credibility. The biological inspiration from synaptic consolidation provides a principled framework for thinking about memory consolidation in RL.

**Weaknesses:**

1. The environments remain relatively simple - continuous mass changes or slip probabilities don't fundamentally alter task structure. Performance changes are often modest and inconsistent across environments.

2. The paper lacks theoretical justification for why SF consolidation should outperform Q-value consolidation. The technical novelty is limited - essentially swapping Q-values with SFs in an existing consolidation mechanism. The choice of noisy sine functions seems arbitrary, and testing only one type of non-stationarity per environment (mass in MuJoCo, action replacement in Four Rooms) limits generalizability claims.

**Questions:**

1. How sensitive are the results to the specific form of continuous non-stationarity? Have you tested with other continuous functions beyond noisy sines?

2. What's the computational overhead of maintaining multiple consolidation variables compared to baseline methods?

3. Can you provide theoretical insight into when SF consolidation should outperform Q-value consolidation?

4. Could you clarify the reparameterization trick used for training with analytically computed consolidation variables?

---

> ### Author Response · Authors · 2025-11-29
>
> Thank you for taking the time to review our rebuttal. We sincerely appreciate your thoughtful comments and are glad to have the opportunity to provide further clarifications. Below, we respond to your excellent points. Please don’t hesitate to reach out if you have any additional questions or concerns.
>
> # 1. Environments remain relatively simple. Continuous mass changes or slip probabilities don't fundamentally alter task structure
> We appreciate the reviewer’s motivation in making this comment. But, the forms of non-stationarity we study in the MuJoCo benchmark and the slippery Four Rooms environment are widely used for evaluating Successor Features and continual RL agents [1–4]. In MuJoCo, changes in mass meaningfully alter the underlying transition dynamics and thereby the optimal policy. In the slippery Four Rooms environment, alternating the reward function produces two distinct optimal policies, which constitutes a genuine change in task structure. Our intention in using these settings is to evaluate methods under realistic and established forms of non-stationarity that directly probe the stability–plasticity trade-off. In contrast, if we radically altered the task structure as the reviewer proposes, then there would be less of a need for balance between stability and plasticity, as strong plasticity would necessarily be required to learn totally new SFs. Respectfully, we would argue that such a test would be contrary to what we are trying to study in this paper.
>
> [1] Touati, Ahmed, Jérémy Rapin, and Yann Ollivier. "Does zero-shot reinforcement learning exist?." arXiv preprint arXiv:2209.14935 (2022).
>
> [2] Dohare, Shibhansh, et al. "Loss of plasticity in deep continual learning." Nature 632.8026 (2024): 768-774.
>
> [3] Chua, Raymond, et al. "Learning successor features the simple way." Advances in Neural Information Processing Systems 37 (2024): 49957-50030.
>
> [4] Xie, Annie, James Harrison, and Chelsea Finn. "Deep reinforcement learning amidst lifelong non-stationarity." arXiv preprint arXiv:2006.10701 (2020).

---

> ### Author Response · Authors · 2025-11-29
>
> # 2. Performance changes are often modest and inconsistent across environments
> We respectfully disagree with the reviewer’s assessment.
>
> Across the environments we study, our method shows clear improvements over all baselines, as evidenced in Figures 1, 3, and 5. These gains are not modest, and they are especially strong in both the MuJoCo and Slippery Four Rooms benchmarks, where our approach consistently outperforms elastic weight consolidation (EWC), plasticity injection (P-last), and continual backprop (CBP), often by a substantial margin. The fact that our method excels precisely in the settings designed to expose stability–plasticity failures further underscores the significance of these improvements. Overall, the results demonstrate robust and meaningful advantages that directly advance continual RL evaluation.

---

> ### Author Response · Authors · 2025-11-29
>
> # 3. How sensitive are the results to the specific form of continuous non-stationarity? Have you tested with other continuous functions beyond noisy sines?
> We thank the reviewer for this thoughtful question. In this work, we focus on (noisy) sine drifts because they are the standard and adopted choice for modelling continuous non-stationarity in prior work [4]. Sine functions provide a simple, well-controlled, and interpretable way to vary the smoothness, magnitude, and timescale of environmental drift, allowing direct comparison with established benchmarks. To assess sensitivity to this choice, we additionally varied the amplitude of the sinusoidal drift across the different embodiments in MuJoCo (Figure 10 in Appendix D). While absolute performance naturally changes with drift difficulty, the results remain consistent, and our consolidation approach consistently outperforms the baselines. These results reflect robustness across varied amplitudes of the noisy sine functions used to simulate continuous non-stationarity.
>
> We are also exploring other forms of continuous drift, such as Ornstein–Uhlenbeck processes, and will report the findings once the results are ready.

---

> > ### Author Response · Authors · 2025-12-04
> >
> > Unfortunately, our compute cluster had technical issues for the last two weeks and we were unable to complete the experiments using  Ornstein–Uhlenbeck processes in time. We will report our approach of using a noisy sine function as a limitation.

---

> ### Author Response · Authors · 2025-11-29
>
> # 4. What's the computational overhead of maintaining multiple consolidation variables compared to baseline methods?
> We thank the reviewer for this helpful suggestion. We agree that reporting the computational complexity of the proposed methods would strengthen the paper. Providing an accurate analysis requires some additional work on our side, and we will include a complexity breakdown in the revised manuscript in the coming days and provide an update here.

---

> > ### Author Response · Authors · 2025-12-04
> >
> > We have now added the computational cost analysis to Section O in the appendix. We evaluated training throughput (FPS) in both the Slippery Four Rooms environment and the Humanoid embodiment in MuJoCo. Overall, we find that the consolidation-augmented model (SF + SC) is slower than the non-consolidated baseline, 26.79% slower than SF in Slippery Four Rooms and 11.50% slower in Humanoid.
> >
> > This slowdown is expected because the consolidation dynamics perform analytic, non-parallelisable updates across the consolidation variables at every training step. These updates are not handled by backpropagation and must be executed sequentially, introducing a constant-factor overhead inherent to the mechanism rather than an implementation inefficiency.
> >
> > Importantly, the overhead remains modest, does not increase with task complexity, and, as anticipated, increases predictably with the number of consolidation variables used. Despite the additional computation, learning is not hindered in any experiment, and the substantial gains in stability and reduced forgetting outweigh this cost.

---

> ### Author Response · Authors · 2025-11-29
>
> # 5. Theoretical insights on when SF consolidation would outperform Q-value consolidation?
> We thank the reviewer for the opportunity to clarify the theoretical differences between consolidating Q-values and consolidating Successor Features (SFs). SFs encode information only about the transition dynamics of the environment, and not the rewards. Consolidating SF parameters stabilizes this predictive structure across time, giving the agent a reliable representation of the underlying dynamics even as policies or reward signals change. Because the task-encoding vector w can be rapidly relearned via simple regression, updating Q-values becomes computationally efficient and highly adaptable.
>
> In contrast, consolidating Q-values entangles transition dynamics and reward information, making the stored values less flexible for reuse across tasks. Thus, SF consolidation better supports continual learning when dynamics shift gradually or rewards change. We note, however, that in settings where the underlying features or dynamics change abruptly and substantially, the advantage of SF consolidation may diminish, as the predictive structure itself requires relearning.

---

> ### Author Response · Authors · 2025-11-29
>
> # 6. Clarification of reparameterization trick used for training
>
> We thank the reviewer for the opportunity to clarify how the reparameterization trick is used in our cross-attention model. The idea is closely related to the reparameterization trick in variational autoencoders (VAEs), but adapted to our multi-timescale SF consolidation setting.
>
> In our architecture, only the fastest-timescale SFs ($SF_{u1}$) are part of the computational graph and are learned directly via the Q-SF-TD loss (Eq. 5) and perturbed using the consolidaiton dynamics (Eq. 11), while all slower-timescale SFs are updated analytically through the consolidation dynamics (Eq. 12) and therefore lie outside the computational graph. To incorporate information from these analytically updated SFs into the learnable $SF_{u1}$, we express $SF_{u1}$ as a deterministic base term plus a set of additive correction terms derived from the differences between successive timescales ($SF_{u2} − SF_{u1}, SF_{u3} − SF_{u2}, …$). These differences are used as keys and values in the cross-attention mechanism, whose output is added to $SF_{u1}$.
>
> This formulation mirrors the reparameterization trick in VAEs, instead of directly backpropagating through analytically updated quantities, we transform them into deterministic additive components that modulate the learnable representation, enabling gradients to propagate through $SF_{u1}$ in a stable and well-defined way. A visual depiction of how this reparameterization operates in our model is provided in Figure 19 in Appendix J.
>
> [5] Kingma, Diederik P., and Max Welling. "Auto-encoding variational bayes." arXiv preprint arXiv:1312.6114 (2013).

---

### Official Review · Reviewer_PeYf · 2025-10-31

**Soundness:** 4
**Presentation:** 3
**Contribution:** 3
**Rating:** 6
**Confidence:** 3

**Summary:**

The paper studies continual RL under continuous (rather than abrupt) non-stationarity and argues that stability is the principal bottleneck, and not plasticity. The authors employ the Successive Features (SF) framework for RL and propose to adapt the neuro-inspired consolidation mechanisms of Benna et al. (2016) to stabilise the feature themselves (as opposed to consolidate the synapses). They show via several experiments that consolidation of SF yields superior performances/learning efficiency than either plasticity-injection models or stabilisation of Q-function/policies in environments that experience continuous changes. Ablation studies highlight the importance of multi-scale stabilisation for performance, while a cross-attention analysis on the use of the SF suggests complementary roles of fast- and slow-consolidation variables.

**Strengths:**

This work represents a welcomed and timely contribution to the field of continuous learning in variable environments. The focus on continuous non-stationarity is a realistic and improves on other techniques that relied on clearly defined task-boundaries. Of particular importance is the result that targeting predictive representations (SFs) for multi-timescale synaptic consolidation under continuous non-stationarity is more effective than stabilizing Q-functions/policies.

In general, the authors provided comparisons with multiple baselines (P-last, CBP, EWC, SC) and metrics (AUC, sample steps), strengthening their case that SF+SC offers excellent performance across the different tasks and environments tested. Furthermore, the paper is well written and executed, with clear figures and explicit questions that clarify the goal of the presented experiments.

**Weaknesses:**

While I believe this work represents a valuable contribution and is already above acceptance level at the current stage, I have listed some weakness/limitations I noticed and I would be happy to raise my score if the author could address them.

1. Author used DDPG as the primary continuous-control baseline. However, DDPG can be less robust to than other actor-critic techniques (TD3, SAC), which may understate the competitiveness of Q- or policy-centric stabilization.
2. The paper argues that SGD is necessary to preserve timescales (Appendix B), but it is unclear whether all compared methods (EWC, CBP, plasticity injections) were trained under exactly the same optimizer/step-size schedules. Fair comparisons require an explicit fixed choice across methods (or provide evidence of robustness against optimizer change).
3. The authors argues that replay-based approaches are problematic because “new vs. old” becomes ill-defined without task boundaries. However, there exists several variants (Kim et al., 2020, Chen & Lin, 2021) that relax such constraints and enable replay-based continual learning without knowledge of explicit task-boundaries. It would have thus be helpful to see how the author's approach stack against such baselines.

----------
### Typos:
- L189: $w$ should be bold

**Questions:**

1. The noisy sinusoidal drift is periodic and potentially predictable. Could this fact be a factor in explaining why preserving SFs is more efficient? Would the situation change if we start to include non-periodic/stochastic drifts?
2. In the experiments consolidation capacity $C_k$ is always fixed to a single choice of scaling, as well as the flow rates $g_{1,2}$. How important are this parameters for the success of the algorithm?
3. Does SF + SC act only on the last layer (SF head), or do you consolidate earlier representation layers as well? If only on the head, is the drift in deeper representations negligible or mitigated indirectly?
4. Since SC is inspired by synaptic consolidation models, have you analyzed whether the timescales learned by SF + SC correspond to biologically plausible values (e.g., logarithmically spaced memory decay constants)?
5. The environments use a noisy sinusoidal drift for dynamics. How sensitive are your conclusions (e.g., superiority of SC over EWC) to the amplitude and frequency of the drift? Is there a regime in which plasticity injection becomes competitive again?
6. Do you have an intuition for *why* SF are better suited for consolidation (maybe lower variance?)?
7. What is the computational cost of SC compared to standard EWC? Are there measurable differences in GPU memory or update latency as K increases?

---

> ### Author Response · Authors · 2025-11-29
>
> # 1.The use of DDPG vs  TD3 or SAC
> We thank the reviewer for bringing up this point.
>
> We chose DDPG because our implementation already incorporates the core TD3 components, which are twin critics and target-policy smoothing. This follows the same design used in prior MuJoCo studies on DDPG and successor features [1, 2, 4]. The only TD3 feature we omit is the delayed policy update, as in our experiments, the agent learns stably without it. Importantly, using this DDPG variant ensures direct comparability with past SF and continual RL work that adopts identical architectures [1,3, 4].
>
> Regarding SAC, prior studies have shown that SAC does not always match or exceed the performance of DDPG/TD3-style deterministic actor–critic agents on certain MuJoCo setups [2, 3]. For these reasons, we opted for a widely adopted and well-understood baseline that aligns with established SF benchmarks while still embodying the essential stabilizing components of modern actor–critic methods.
>
> [1] Chua, Raymond, et al. "Learning successor features the simple way." Advances in Neural Information Processing Systems 37 (2024): 49957-50030.
>
> [2] Fujimoto, Scott, Herke Hoof, and David Meger. "Addressing function approximation error in actor-critic methods." International conference on machine learning. PMLR, 2018.
>
> [3]  Yarats, Denis, et al. "Mastering visual continuous control: Improved data-augmented reinforcement learning." arXiv preprint arXiv:2107.09645 (2021).
>
> [4] Touati, Ahmed, Jérémy Rapin, and Yann Ollivier. "Does zero-shot reinforcement learning exist?." arXiv preprint arXiv:2209.14935 (2022).

---

> > ### Author Response · Authors · 2025-11-29
> >
> > # 6. The noisy sinusoidal drift is periodic and may be predictable
> > We thank the reviewer for providing this comment. While the sinusoidal drift is periodic and may be predictable, we chose to use the sine function as it aligns with prior work that studied continuous changes [5]. We are also exploring other forms of continuous drift, such as Ornstein–Uhlenbeck processes, and will report the findings if the results are ready before the end of discussion period.
> >
> > [5] Xie, Annie, James Harrison, and Chelsea Finn. "Deep reinforcement learning amidst lifelong non-stationarity." arXiv preprint arXiv:2006.10701 (2020).

---

> > > ### Author Response · Authors · 2025-12-04
> > >
> > > Unfortunately, our compute cluster had technical issues for the last two weeks and we were unable to complete the experiments using  Ornstein–Uhlenbeck processes in time. We will report our approach of using a noisy sine function as a limitation.

---

> > ### Author Response · Authors · 2025-11-29
> >
> > # 8. Is the consolidation applied on the last layer or the whole network which includes the representation layers as well?
> >
> > We thank the reviewer for the question.
> >
> > We apply consolidation to the whole network which includes the representations layers. This also aligns with previous approaches of applying consolidation to value functions [6] and policies [7].
> >
> > [6] Kaplanis, Christos, Murray Shanahan, and Claudia Clopath. "Continual reinforcement learning with complex synapses." International Conference on Machine Learning. PMLR, 2018.
> >
> > [7] Kaplanis, Christos, Murray Shanahan, and Claudia Clopath. "Policy consolidation for continual reinforcement learning." arXiv preprint arXiv:1902.00255 (2019).

---

> ### Author Response · Authors · 2025-11-29
>
> # 2.Choice of optimizers used in EWC, CBP, plasticity injections
> We thank the reviewer for raising this point and giving us the opportunity to clarify the choice of optimizers.
> All trainable networks in our models, including the actor, critic, and task-encoding vector are optimized using Adam. The only exception is the synaptic consolidation system, whose parameters of the SF variables and Q-value functions evolve according to the analytically defined update dynamics in Equations 11–13. These equations specify a particular form of parameter flow across timescales that must be implemented exactly as written for the consolidation mechanism to exhibit the intended behavior.
> For this reason, the consolidation updates are applied using a simple SGD-style rule (i.e., direct parameter updates without momentum or adaptive moment estimates). Momentum-based optimizers such as Adam would alter the effective update direction and magnitude, thereby distorting the mathematically defined flow dynamics and breaking the intended separation of timescales. As we proved mathematically in Appendix B, adding momentum indeed disrupts the preservation of the prescribed timescales.
> We also emphasize that the learning rates across all models, including EWC, CBP, and plasticity-injection, were kept consistent for fair comparison. Adjusting the learning rate for the models did not yield improved performance.

---

> > ### Author Response · Authors · 2025-11-29
> >
> > # 7. Importance of capacity and flow rate parameters
> >  We thank the reviewer for the question. It is important for the capacity and the flow rate to be set as they are mathematically defined in the dynamical system to ensure that they obey the power law, and operate at different timescales. If the capacity and flow rate differs from the mathematical definitions, then the balance between plasticity and plasticity will no longer be maintained.

---

> ### Author Response · Authors · 2025-11-29
>
> # 3. Comparison with replay based methods
> We thank the reviewer for pointing this out. We would like to emphasise that in our Mujoco benchmark, there is no task identity, no task boundary, and no discrete task switch. The environment is a single continuous control task whose underlying dynamics drift smoothly over time. As a result, the agent’s replay buffer naturally contains a continuously evolving mixture of recent and older transitions.

---

> ### Author Response · Authors · 2025-11-29
>
> # 4.Typo for L189: w
> We thank the reviewer for pointing this out and we will make the correction in the revised manuscript.

---

> ### Author Response · Authors · 2025-11-29
>
> # 5.References for replay methods cannot be found
> We thank the reviewer for bringing our attention to the potential references but we could not find them: (Kim et al., 2020, Chen & Lin, 2021). Can the reviewer kindly share the full title of the papers?

---

> ### Author Response · Authors · 2025-11-29
>
> # 9. Do the timescales learned by SF + SC correspond to biologically plausible values?
> We thank the reviewer for the thoughtful question. The timescales induced by the Benna–Fusi consolidation dynamics are inspired by, but not intended to match one-to-one, biological values. The logarithmically spaced decay constants produced by the cascade model are considered biologically plausible in the sense that they mirror the wide hierarchy of synaptic and systems consolidation timescales observed in neuroscience, from seconds and minutes (e.g., early LTP) to hours, days, and even months (e.g., late LTP, protein synthesis–dependent consolidation, systems consolidation). Our implementation preserves this qualitative structure. However, we do not claim that the exact numerical timescales directly correspond to measured biological rates.  Rather, they capture the underlying principle that memory traces evolve on multiple interacting timescales with approximately power-law decay.

---

> ### Author Response · Authors · 2025-11-29
>
> # 10. Sensitivity to Drift Amplitude and Frequency
> We thank the reviewer for this helpful question. To assess sensitivity to drift magnitude, we systematically varied the amplitude of the noisy sinusoidal drift across different MuJoCo embodiments (see Figure 10 in Appendix D). Across these settings, our consolidation model consistently outperforms EWC (Figure 16 in Appendix H), indicating robustness to a wide range of drift intensities. For each embodiment, we also explored the maximum amplitude that can be applied before the underlying physics engine becomes unstable; within all feasible ranges, our conclusions remain unchanged. While we focused primarily on amplitude (the more impactful parameter for MuJoCo dynamics), the frequency is fixed to ensure consistency across benchmarks, and we will clarify this choice in the revised manuscript.
>
> If the frequency is reduced, we would expect Continual Backprop (CBP) to be able to adapt and maintain reasonable performance in MuJoCo. However, plasticity-injection methods remain fundamentally brittle even when the drift becomes slow. Injecting plasticity into the critic of an actor-critic agent is a drastic intervention because the actor depends on a stable critic to provide meaningful gradients. Even small resets disrupt this coupling and lead to persistent drops in learning performance. Thus, while lower frequencies reduce the overall difficulty of the non-stationarity, we would expect plasticity injection continues to underperform, whereas CBP and our consolidation approach remain effective.

---

> ### Author Response · Authors · 2025-11-29
>
> # 11. Is there a regime in which plasticity injection becomes competitive again?
> We thank the reviewer for this thoughtful question. Our results indicate that plasticity injection can become competitive only in a particular regime, and this depends strongly on the environment and architecture. In MuJoCo (Figures 1 and 5), resetting even just the last layer of the critic is highly disruptive for actor–critic methods, as the actor relies on a stable critic, hence injection consistently harms learning. However, in the 3D Four Rooms environment (Figure 3a), plasticity injection on the last layer (red) does improve re-learning efficiency during the second exposure, suggesting that it can be beneficial when the model is of Deep Q-Network like, using discrete actions, and when the critic reset does not destabilize the policy. Overall, these findings suggest that plasticity injection is not a robust mitigation strategy for continuous-control actor–critic systems like DDPG. In such situations, continual backpropagation (CBP) offers a more reliable alternative.

---

> ### Author Response · Authors · 2025-11-29
>
> # 12. Computational cost of SC compared to standard EWC
> We thank the reviewer for this question. We will report the computational complexity of the proposed methods. Providing an accurate analysis requires some additional work on our side, and we will include a complexity breakdown in the revised manuscript in the coming days and provide an update here.

---

> > ### Author Response · Authors · 2025-12-04
> >
> > We have now added the computational cost analysis to Section O in the appendix. We evaluated training throughput (FPS) in both the Slippery Four Rooms environment and the Humanoid embodiment in MuJoCo. Overall, we find that the consolidation-augmented model (SF + SC) is slower than the non-consolidated baseline, 26.79% slower than SF in Slippery Four Rooms and 11.50% slower in Humanoid.
> >
> > This slowdown is expected because the consolidation dynamics perform analytic, non-parallelisable updates across the consolidation variables at every training step. These updates are not handled by backpropagation and must be executed sequentially, introducing a constant-factor overhead inherent to the mechanism rather than an implementation inefficiency.
> >
> > Importantly, the overhead remains modest, does not increase with task complexity, and, as anticipated, increases predictably with the number of consolidation variables used. Despite the additional computation, learning is not hindered in any experiment, and the substantial gains in stability and reduced forgetting outweigh this cost.

---

> ### Author Response · Authors · 2025-11-29
>
> # 13. Are there measurable differences in GPU memory or update latency as K increases?
> We thank the reviewer for this question. Similar to the point made above in (12) on computational cost, we will report the computational complexity of the proposed methods. Providing an accurate analysis requires some additional work on our side, and we will include a complexity breakdown in the revised manuscript in the coming days and provide an update here.

---

### Official Review · Reviewer_DS9m · 2025-11-04

**Soundness:** 2
**Presentation:** 2
**Contribution:** 2
**Rating:** 4
**Confidence:** 3

**Summary:**

This paper investigates the stability-plasticity tradeoff in deep RL under continuous non-stationary environments. The authors modify existing benchmarks (3D Miniworld Four Rooms, MuJoCo) to incorporate gradual dynamics changes via noisy sine functions. They find that stability (via synaptic consolidation) is more critical than plasticity (via parameter resets), and propose consolidating Successor Features (SFs) across multiple timescales. The approach outperforms consolidating Q-values and plasticity-focused methods. A cross-attention analysis reveals that faster timescales contribute more to learning, though this mechanism only improves simpler embodiments.

**Strengths:**

- Continuous non-stationarity is a more realistic and understudied problem formulation compared to discrete task boundaries commonly used in continual learning; the gradual dynamics changes via noisy sine functions better reflect real-world scenarios
- Comprehensive experimental validation across two environments and multiple embodiments consistently demonstrates SF+SC superiority over both Q-value consolidation and plasticity injection methods

**Weaknesses:**

- Limited technical novelty as the work primarily combines two existing techniques (Successor Features and synaptic consolidation); the contribution is largely empirical without new algorithmic insights
- Experiments confined to relatively simple domains with state-based observations; scalability to high-dimensional spaces (pixel-based observations, complex robotics) remains unclear and untested

**Questions:**

- Can you test on at least one high-dimensional domain (e.g., pixel-based Atari with continuous dynamics changes) to demonstrate the approach scales beyond state-based control?
- Why does the cross-attention mechanism fail for complex embodiments (Humanoid, Quadruped)?

---

> ### Author Response · Authors · 2025-11-29
>
> # 1. Limited technical novelty
> We respectfully disagree with the reviewer’s assessment that this work does not provide new algorithmic insights.
>
> First, while our framework builds upon the established concepts of Successor Features (SFs) and synaptic consolidation, these mechanisms have not been previously studied together at the level of predictive representations. Prior work applying synaptic consolidation has focused on preserving value functions or policies, which differ fundamentally from predictive representations. Consolidation at the level of representation remains largely unexplored in reinforcement learning, and our formulation introduces a new mechanism for stabilising predictive structure across timescales.
>
> Second, our ablation studies on the number of consolidation variables (Figure 7) reveal the importance of learning multiple SFs across various timescales in continuous non-stationary settings. We find that incorporating more slower timescale SFs consistently improves learning efficiency, highlighting a general principle: distributing consolidation across a spectrum of slower timescales enhances stability and adaptability.
>
> Third, beyond combining SFs and synaptic consolidation, we introduce a cross-attention mechanism that operates directly on the SFs learned at multiple timescales. In this model, the task-encoding vector acts as a query, while the mult-timescales SFs act as keys and values. This design enables the agent to dynamically retrieve predictive representations most relevant to the current task, effectively turning the consolidated SFs into a contextual memory system.
>
> To the best of our knowledge, prior work employing attention in the context of SFs has only been applied to the output of the encoder and not on the SFs themselves. Moreover, by analysing the resulting attention weights, we quantify the relative contribution of each timescale to decision-making, thus providing algorithmic and interpretable insights into how the multi-timescale SFs interact during continual learning.
>
> [1] Kaplanis, Christos, Murray Shanahan, and Claudia Clopath. "Continual reinforcement learning with complex synapses." International Conference on Machine Learning. PMLR, 2018.
>
> [2] Kaplanis, Christos, Murray Shanahan, and Claudia Clopath. "Policy consolidation for continual reinforcement learning." arXiv preprint arXiv:1902.00255 (2019).
>
> [3] Carvalho, Wilka, et al. "Composing task knowledge with modular successor feature approximators." arXiv preprint arXiv:2301.12305 (2023).

---

> ### Author Response · Authors · 2025-11-29
>
> # 2. Experiments confined to relatively simple domains with state-based observations
> We would like to clarify that our experiments are not limited to simple state-based domains. In addition to the MuJoCo Benchmarks, we included a 3D Four Rooms environment with egocentric pixel observations (see Figure 4). This setting poses a more challenging task, involving sparse rewards and discrete actions from high-dimensional visual inputs. As shown in Figure 3, our model (SF +SC) demonstrates improved learning efficiency, requiring fewer steps to learn the optimal policy compared to baseline approaches.

---

> ### Author Response · Authors · 2025-11-29
>
> # 3. Scalability to high-dimensional spaces remains unclear and untested
> We agree with the reviewer that testing the technique in high-dimensional spaces is important. But, we would point out that our experiments include high-dimensional continuous control environments in MuJoCo, covering multiple embodiments such as Humanoid, which has 67-dimensional state space and 21-dimensional action space, making it one of the most complex and challenging benchmarks in reinforcement learning. These environments provide rich continuous and non-stationary conditions, directly aligning with our study’s focus on adaptation under continuous changing dynamics. While we are not aware of other standardized robotic domains that offer higher complexity and similarly allow for continuous dynamics changes, we would welcome further clarification or suggestions of such benchmarks that could complement our study.

---

> ### Author Response · Authors · 2025-11-29
>
> # 4.Why does the cross-attention mechanism fail for complex embodiments (Humanoid, Quadruped)?
> We respectfully clarify that the cross-attention mechanism did not fail for complex embodiments such as Humanoid and Quadruped, as shown in Figure 22 in Appendix K. Rather, its learning progress is slower in these settings due to the higher computational complexity of learning attention over larger state and action spaces. Importantly, even in these complex embodiments, the cross-attention model still outperforms the baseline SF model without synaptic consolidation. This further highlights the benefit of learning SFs across multiple timescales, which enables more robust predictive representations under challenging, high-dimensional conditions.

---

> ### Author Response · Authors · 2025-11-29
>
> Thank you for taking the time to review our rebuttal. We sincerely appreciate your thoughtful comments and are glad to have the opportunity to provide further clarifications. Below, we respond to your excellent points. Please don’t hesitate to reach out if you have any additional questions or concerns.

---

### Meta-Review · Area_Chair_GuKd · 2026-01-07

**Summary:**

The paper investigates continual reinforcement learning under gradual non-stationarity and argues that stability, rather than plasticity, is the dominant limitation. The authors propose applying multi-timescale synaptic consolidation to successor features instead of Q-values, demonstrating consistent improvements across MuJoCo and 3D Four Rooms environments. Reviewers appreciated the realistic problem setting and experimental breadth, but raised concerns about limited algorithmic novelty, lack of explicit stability–plasticity metrics, performance inconsistency across environments, and reliance on sinusoidal drift. The rebuttal clarified several methodological points and added a computational cost analysis, though key empirical extensions (e.g., OU drift, quantitative stability/plasticity disentangling) remain incomplete. The reviewer scores and remaining concerns do not provide sufficient support for acceptance.

**Reviewer Concerns:**

### Addressed by Rebuttal
- **Evidence beyond low-dimensional/state-only settings (Reviewer DS9m)**: The authors clarified that the 3D Four Rooms experiments include egocentric pixel observations, and that MuJoCo evaluation spans multiple embodiments including high-dimensional ones (e.g., Humanoid). This addresses the concern that the study was confined to simple state-based domains.
- **Cross-attention interpretation and alleged failure in complex embodiments (Reviewer DS9m)**: The rebuttal clarifies that cross-attention learning is slower in complex embodiments but remains beneficial relative to the SF baseline, aligning the mechanism discussion with the reported results.
- **Baseline/optimizer fairness (Reviewer PeYf)**: The authors clarified that their DDPG implementation already includes key TD3 components. They further noted that all trainable networks are optimized with Adam using consistent learning-rate schedules. Finally, the consolidation updates follow analytically defined dynamics implemented as direct parameter flows. This resolves the main concerns on fair comparisons with the baselines.
- **Computational overhead (Reviewers an5Z, PeYf, P8uF)**: The authors reported that SF+consolidation is slower than SF alone, which enables a clearer practicality trade-off assessment.
- **Presentation/clarifications (Reviewer P8uF)**: The authors disambiguated terminology (“continuous” → “naturalistic continual”), explained why plasticity injection destabilizes actor–critic training, and acknowledged a plotting/legend issue to be fixed.

### Still Outstanding:
- **Strength of the “stability > plasticity” conclusion without explicit metrics (Reviewer P8uF)**: The paper primarily infers stability/plasticity bottlenecks from which family of methods performs best. The rebuttal sketches possible metrics/analyses but does not deliver quantitative disentangling (and later notes compute constraints).
- **Limited algorithmic novelty / deeper theory (Reviewers DS9m, an5Z)**: Consolidating SFs rather than Q-values is well motivated and empirically supported. However, the core advance is still largely a principled integration with extensive experiments. The lack of a  sharper theoretical characterization of when SF consolidation should dominate limits the strength of the contribution.
- **Generality of the non-stationarity model (Reviewers PeYf, an5Z)**: Most evidence relies on noisy sinusoidal drift. Additional drift processes (e.g., OU/stochastic or non-periodic) were not completed.
- **Positioning vs additional baselines and statistics (Reviewers PeYf, P8uF)**: Direct comparisons to replay-based continual RL methods without task boundaries are missing (the authors requested full citations from PeYf but did not receive them). In addition, requests for stronger statistical testing/significance reporting are not fully addressed.

**Reviewer Scores:**

- **Reviewer DS9m: 4 → 4.** The main requests (evidence beyond simple/state-only settings and clarification about the cross-attention results) were directly addressed, but the reviewer's primary concern about limited technical novelty was not resolved by the rebuttal and would likely keep the score.
- **Reviewer PeYf: 6 → 6.** The principal fairness/optimization questions and computational-overhead request were addressed; remaining questions about drift predictability and missing comparisons might keep the reviewer at 6.
- **Reviewer an5Z: 6 → 6.** The rebuttal adds clarity on motivation, robustness (amplitude variation), and overhead, but the reviewer's concerns on novelty and theoretical evidence are only partially addressed, so a stable score is most plausible.
- **Reviewer P8uF: 4 → 4.** Several confusions and omissions (terminology, baseline rationale, computational cost, and a figure issue) were addressed, but the reviewer's main request for quantitative stability/plasticity metrics was not delivered. In addition, the authors did not address the request on the statistical testing, which would likely keep the reviewer at their original score.

---

### Decision · Program_Chairs · 2026-01-26

Reject